# Disentangling Behaviorally Relevant Latent Dynamics in Multimodal Neural Time-series

## Abstract

Multimodal neural data can enable a more complete understanding of brain dynamics underlying behavior. Modalities such as neuronal spiking activity, local field potentials (LFPs), and behavioral signals capture diverse spatiotemporal aspects of brain processes. By leveraging these complementary strengths, multimodal neural fusion can provide a unified, rich representation of brain-behavior processes and address the limitations of single-modality analyses, such as incomplete or noisy data. While recent works have jointly modeled behavior and neural data to disentangle sources of variability, they largely rely on latent variable models that use a single modality of neural data. Here we develop a nonlinear dynamical model, termed BREM-NET, that integrates behavioral signals and multiple neural modalities—such as LFPs and spike counts—with distinct statistical characteristics and temporal resolutions into a unified framework, and performs multimodal neural fusion during inference. In two independent public multimodal neural datasets, we show that BREM-NET nonlinearly fuses information across neural modalities while also disentangling behaviorally relevant multimodal neural dynamics. Doing so results in inferring more accurate disentangled latent dynamics, as reflected in enhanced behavior decoding and neural prediction compared to multimodal baselines. Furthermore, BREM-NET enables disentanglement and multimodal fusion even when different neural time-series modalities are asynchronous and have distinct temporal resolutions, which is a major challenge in real-world neural recordings. This framework provides a new tool for studying behaviorally relevant neural computations across different spatiotemporal scales of brain activity.

## 1 Introduction

Understanding the dynamics of brain activity across different spatiotemporal scales and their relationship to behavior is a fundamental challenge in neuroscience. These different scales can be captured via different modalities. Neuronal spike trains, local field potentials (LFPs), and behavioral signals each offer distinct yet complementary insights into brain function (Pesaran et al., 2002; Penttonen & Buzsáki, 2003; Belitski et al., 2010; Einevoll et al., 2013; Stavisky et al., 2015; Pesaran et al., 2018; Lu et al., 2021; Ramezani et al., 2024). Spikes capture the spatial scale of neurons, reflect rapid transient neural events at a millisecond timescale, and are often modeled with Poisson distributions due to their discrete nature. LFPs measure a larger spatial scale and slower, sustained oscillatory processes that evolve over longer timescales than spikes, and are typically modeled with Gaussian distributions (Buzsáki et al., 2012; Hsieh et al., 2018; Ahmadi et al., 2021). Behavioral signals represent high-level outputs of the nervous system, introducing an additional layer of difficulty due to their complex relationships with neural activity (Gallego et al., 2020; Urai et al., 2022).

Given the above complementary aspects, enabling the integration of these modalities can provide a powerful, unified dynamical modeling framework for uncovering the multimodal neural mechanisms driving behavior. If developed, such models can also improve neural-behavioral predictions, particularly for brain-computer interfaces (BCIs). However, developing such models remains challenging for several reasons. First, neural modalities have both *behaviorally relevant* dynamics that are shared with behavior dynamics, and other neural dynamics, which we refer to as *neural-specific* dynamics. Disentangling these two types of dynamics is important for understanding neural-behavior relationships. Second, different modalities such as discrete spikes and continuous LFPs have distinct statistical and temporal characteristics. Third, in real-world recordings, neural modalities are often

asynchronous—that is, they are sampled at different temporal resolutions—or suffer from dropouts due to noise or electrode failures, complicating their multimodal integration. Finally, the dynamics in these modalities can be complex and nonlinear. Despite recent advances in multimodal frameworks, existing approaches do not simultaneously incorporate behavior and multiple asynchronous neural modalities, such as spikes and LFPs, into a unified dynamical model (Ramezani et al., 2024).

Indeed, one line of work has made notable advances in modeling a single modality of neural activity jointly with behavior to learn behaviorally relevant neural dynamics with latent-variable modeling Sani et al. (2021); Hurwitz et al. (2021); Schneider et al. (2023); Gondur et al. (2023); Sani et al. (2024), Zhang et al. (2025); Schulz et al. (2025). This body of work has shown that disentangling behaviorally relevant neural dynamics enables interpretable analyses of how neural activity relates to behavior, allowing for new neuroscientific insights. However, these methods are constrained to using a single neural modality, such as spikes alone or LFP alone, and do not perform multimodal neural fusion. Thus, a major unaddressed challenge in neural-behavioral models is to simultaneously capture the diverse spatiotemporal neural dynamics of behavior, at both the fine scale of spikes and the larger network-level scale of LFP.

Another line of work has developed multimodal models to aggregate information across different time-series, but without separating their shared and distinct dynamics. Thus, while these approaches have been used to fuse information across multiple neural modalities, they do not disentangle the behaviorally relevant neural dynamics of such modalities (Coleman et al., 2011; Zhou & Wei, 2020; Abbaspourazad et al., 2021; Singh Alvarado et al., 2021; Kramer et al., 2021; Rezaei et al., 2023; Ahmadipour et al., 2024). This can conflate behaviorally relevant neural dynamics with other neural-specific dynamics. Addressing this challenge in multimodal models is important for neuroscientific studies of how multimodal neural activity relates to behavior and for neural-behavioral prediction.

**Contributions** To address this gap, we propose **B**ehaviorally **Re**levant modeling of **M**ultimodal **Ne**ural **T**ime-series (**BREM-NET**), a novel nonlinear dynamical model that simultaneously achieves the following capabilities: 1) It jointly incorporates multiple neural modalities with behavioral signals into a unified model. 2) It disentangles behaviorally relevant neural dynamics and neural-specific dynamics in multimodal neural time-series. 3) It enables nonlinear multimodal neural fusion under heterogeneous statistical distributions, 4) It enables integrating modalities with different temporal resolutions or dropouts. We validate BREM-NET on two public non-human primate (NHP) spike-LFP datasets from different brain regions and during different tasks (Flint et al., 2012; O'Doherty et al., 2020). We find that BREM-NET successfully fuses information across neural modalities—even when they are sampled asynchronously—while also disentangling behaviorally relevant neural dynamics. This disentangled learning improves behavior decoding and multimodal neural prediction compared to multimodal baselines. By modeling different spatiotemporal neural scales along with behavior, our approach provides a new tool for investigating multiscale neural computations that underlie behavior in neuroscience and for advancing BCIs by making behavior decoding more accurate and robust via multimodal neural fusion under realistic, asynchronous conditions.

## 2 RELATED WORK

Prior related works fall into two broad categories: (i) models that disentangle behaviorally relevant neural dynamics but just for a single neural modality without enabling multimodal neural fusion, and (ii) models that fuse information across multiple neural modalities but do not disentangle behaviorally relevant neural dynamics. BREM-NET achieves both these capabilities: it not only fuses information across multiple neural modalities even in asynchronous scenarios, but also disentangles their behaviorally relevant neural dynamics.

The first category includes both linear models (Kobak et al., 2016; Sani et al., 2021; Vahidi et al., 2024) and nonlinear deep learning models Zhou & Wei (2020); Hurwitz et al. (2021), Whiteway et al. (2021), Schneider et al. (2023); Gondur et al. (2023); Sani et al. (2024), Wang et al. (2024); Wu et al. (2025) for joint neural-behavioral modeling. Among these works, some recent methods include TNDM (Hurwitz et al., 2021), DPAD (Sani et al., 2024), CEBRA (Schneider et al., 2023), and MMGPVAE (Gondur et al., 2023). TNDM is a sequential autoencoder (similar to LFADS in (Pandarinath et al., 2018)) that optimizes the reconstruction of behavioral data and single-modal Poisson neural data. DPAD uses an RNN-based dynamical model with separate latent states for behaviorally relevant and irrelevant neural dynamics for single-modal Gaussian neural activity. Both

TNDM and DPAD perform inference from one neural modality and thus do not support multimodal neural fusion. CEBRA is a convolutional encoder with a contrastive loss on behavior to extract behaviorally relevant embeddings (Appendix A.2.7). MMGPVAE captures the temporal structure through Gaussian process priors (Casale et al., 2018) and optimizes the evidence-lower bound (ELBO) with variational inference (Appendix A.2.9). However, neither of these methods is designed for handling behavior and multiple neural time-series simultaneously, which our method enables even when neural modalities exhibit distinct temporal resolutions. Nevertheless, we compare to both of them by taking their inputs during inference to be the two neural modalities.

The second category spans both linear and nonlinear models that fuse information across multiple neural modalities. Linear approaches include multiscale dynamical systems (Coleman et al., 2011; Abbaspourazad et al., 2021; Rezaei et al., 2023) and a subspace identification method named MSID (Ahmadipour et al., 2024), which we compare to (Appendix A.2.5). Also, a nonlinear method, termed mmPLRNN Kramer et al. (2021), uses piecewise linear dynamics within a variational autoencoder and optimizes the ELBO, which we compare to (Appendix A.2.8). However, these methods do not disentangle behaviorally relevant neural dynamics. Further, current nonlinear multimodal methods are designed for fusing modalities with the same (rather than distinct) temporal resolutions.

## 3 METHODS

The architecture of BREM-NET, depicted in Fig. 1, is designed for joint modeling of behavior and multiple neural modalities observed simultaneously. The motivation behind our design is to explicitly address two key challenges in multimodal neural modeling: 1) disentangling the behaviorally relevant neural dynamics from neural-specific dynamics unrelated to behavior, and 2) handling asynchronous modalities. To enable disentanglement, BREM-NET employs two RNNs thought a three-stage learning process: to describe behaviorally relevant neural dynamics, the first RNN learns behaviorally relevant neural latents in Stage 1, whose mapping to neural modalities is learned in Stage 2. The second RNN learns neural-specific latents in Stage 3 to capture neural dynamics unexplained by the first RNN latents. To handle asynchronous neural modalities and/or modality dropouts, we introduce separate modality-specific encoders that allow neural modalities to be processed at their native temporal resolutions and that handle modality-specific dropouts.

We model multimodal neural and behavior time-series as observations of a nonlinear dynamical system (Eq.1), whose evolution is captured with latent states $\mathbf{X} = \{\mathbf{x}_k : \mathbf{x}_k \in \mathbb{R}^{n_x}, k \in \mathcal{K}\}$, where $k$ is the time index and $\mathcal{K} = \{1, \ldots, \mathbf{K}\}$. Specifically, we infer the latent states by fusing information across two neural time-series which can have distinct temporal resolutions: $\mathbf{S} = \{\mathbf{s}_k : \mathbf{s}_k \in \mathbb{R}^{n_s}, k = 1, \ldots, K\}$, $\mathbf{Y} = \{\mathbf{y}_t : \mathbf{y}_t \in \mathbb{R}^{n_y}, t \in \mathcal{T}\}$ with $\mathcal{T} = \{1, \ldots, \mathbf{T}\}$ and $\mathcal{T} \subseteq \mathcal{K}$. These distinct index sets $(k, t)$ emphasize that neural modalities can be sampled at different rates, with $k$ indexing the faster sampling rate. Our goal, however, is to perform latent state inference on the timescale of the faster modality, thereby enabling more flexible behavior decoding. In addition to neural observations, during learning, BREM-NET incorporates $\mathbf{Z} = \{\mathbf{z}_k : \mathbf{z}_k \in \mathbb{R}^{n_z}, k = 1, \ldots, K\}$, a behavioral time-series that can be decoded from the latent states. We disentangle the latent states $\mathbf{X}$ into two distinct sets: behaviorally relevant neural latents $\mathbf{X}^{(1)} = \{\mathbf{x}_k^{(1)} : \mathbf{x}_k^{(1)} \in \mathbb{R}^{n_1}\}$ and neural-specific latents $\mathbf{X}^{(2)} = \{\mathbf{x}_k^{(2)} : \mathbf{x}_k^{(2)} \in \mathbb{R}^{n_x - n_1}\}$.

### 3.1 MODEL FORMULATION

We assume that the multimodal neural and behavioral time-series are generated from a nonlinear latent dynamical system described by:

$$
\begin{aligned}
\mathbf{x}_{k+1} &= F(\mathbf{x}_k) + \mathbf{w}_k \\
\mathbf{y}_t \mid \mathbf{x}_t &\sim p_{\mathbf{y}}\big(\mathbf{y}_t \mid \bar{\boldsymbol{C}}_{\mathbf{y}}(\mathbf{x}_t)\big) \\
\mathbf{s}_k \mid \mathbf{x}_k &\sim p_{\mathbf{s}}\big(\mathbf{s}_k \mid \bar{\boldsymbol{C}}_{\mathbf{s}}(\mathbf{x}_k)\big) \\
\mathbf{z}_k \mid \mathbf{x}_k &\sim p_{\mathbf{z}}\big(\mathbf{z}_k \mid \bar{\boldsymbol{C}}_{\mathbf{z}}(\mathbf{x}_k)\big)
\end{aligned}
\tag{1}
$$

In this system, the latent state evolves in time through a recursion function $F(\cdot)$ with additive noise $\mathbf{w_k}$. The latent state is connected to three types of time-series through different distributions $p_y\big(\mathbf{y}_t \mid \bar{\boldsymbol{C}}_{\mathbf{y}}(\mathbf{x}_t)\big), p_s\big(\mathbf{s}_k \mid \bar{\boldsymbol{C}}_{\mathbf{s}}(\mathbf{x}_k)\big)$, and $p_z\big(\mathbf{z}_k \mid \bar{\boldsymbol{C}}_{\mathbf{z}}(\mathbf{x}_k)\big)$ that describe the likelihood distributions of

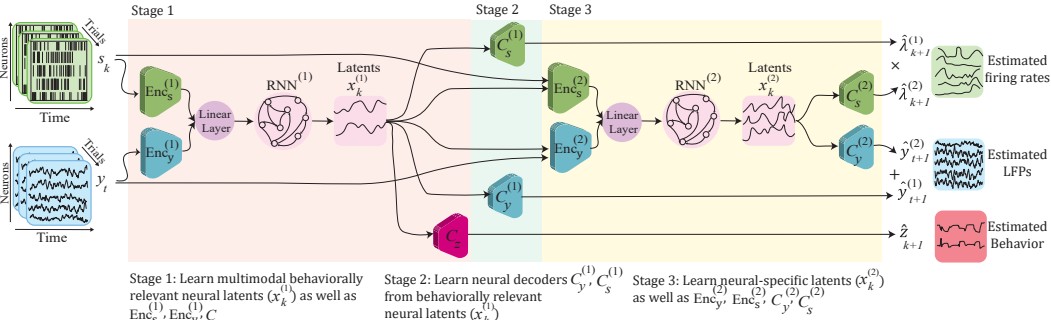

Stage 1: Learn multimodal behaviorally relevant neural latents $(x_k^{(1)})$ as well as $Enc_s^{(1)}, Enc_y^{(1)}, C_z$

Stage 2: Learn neural decoders $C_y^{(1)}, C_s^{(1)}$ from behaviorally relevant neural latents $(x_k^{(1)})$

Stage 3: Learn neural-specific latents $(x_k^{(2)})$ as well as $Enc_y^{(2)}, Enc_s^{(2)}, C_y^{(2)}, C_s^{(2)}$

Figure 1: **BREM-NET model architecture.** BREM-NET's computation graph consists of three stages that take in multimodal neural activities at the current time step and fuse them to predict behavior and multimodal neural activities for the next time step. Stage 1 fuses neural modalities to learn behaviorally relevant neural latents, $\mathbf{x}_k^{(1)}$, by jointly fitting an RNN with modality-specific encoders and a behavior decoder to minimize the negative log-likelihood (NLL) of behavior given the RNN latents, $\mathbf{x}_k^{(1)}$. Stage 2 fits two modality-specific neural decoders to minimize the NLL of multimodal neural time-series given $\mathbf{x}_k^{(1)}$. Stage 3 fuses neural modalities to learn neural-specific latents, $\mathbf{x}_k^{(2)}$, by jointly fitting another RNN with two modality-specific encoders and two modality-specific decoders for multimodal neural data. This stage takes multimodal neural time-series and the already-learned $\mathbf{x}_k^{(1)}$ and minimizes the mixed multimodal NLL of spike-LFP modalities given $\mathbf{x}_k^{(2)}$ and $\mathbf{x}_k^{(1)}$, allowing it to learn residual multimodal neural dynamics that are not predicted by $\mathbf{x}_k^{(1)}$.

distinct neural time-series modalities and behavior time-series respectively, with $\bar{C}_{\mathbf{y}}(\cdot), \bar{C}_{\mathbf{s}}(\cdot), \bar{C}_{\mathbf{z}}(\cdot)$ being nonlinear decoder networks. These distributions can take forms such as Poisson for spike counts, Gaussian for continuous signals, and Bernoulli for binary outcomes.

We nonlinearly fuse multimodal neural time-series $\mathbf{S}$ and $\mathbf{Y}$ with distinct temporal resolutions to infer the latent states using an RNN:

$$\mathbf{x}_{k+1} = A(\mathbf{x}_k) + \mathbf{Enc_y}(\mathbf{y}_t) + \mathbf{Enc_s}(\mathbf{s}_k) \tag{2}$$

where $A(\cdot)$ captures the temporal evolution of latent states, and $\mathbf{Enc_y}(\cdot)$ and $\mathbf{Enc_s}(\cdot)$ are modality-specific nonlinear encoder networks. This nonlinear fusion architecture enables BREM-NET to combine complementary information across distinct neural modalities in a shared latent space, supporting more expressive latent dynamics. In addition to this nonlinear fusion of neural modalities, BREM-NET explicitly addresses the challenge of different temporal resolutions and modality-specific dropouts, which are common in real-world neural recordings; this is achieved via the separate modality-specific encoders $\mathbf{Enc_y}(\cdot)$ and $\mathbf{Enc_s}(\cdot)$. If one modality is unavailable at a given $k$, the corresponding encoder output can be set to zero in Eq. 2, so the state update relies only on any available modality.

In addition to multimodal fusion, another major goal of BREM-NET is to disentangle the behaviorally relevant and neural-specific dynamics within the multimodal neural time-series. We define two sets of latent states (Sani et al., 2021; Hurwitz et al., 2021; Sani et al., 2024): (1) behaviorally relevant neural latents $\mathbf{x}_k^{(1)} \in \mathbb{R}^{n_1}$, which capture the shared dynamics between behavior and multimodal neural time-series, and (2) neural-specific latent states $\mathbf{x}_k^{(2)} \in \mathbb{R}^{n_x - n_1}$, which encode the residual neural dynamics independent of behavior (see Appendix A.1.2 for details). We then expand our formulation in a disentangled form as follows,

$$\begin{bmatrix} \mathbf{x}_{k+1}^{(1)} \\ \mathbf{x}_{k+1}^{(2)} \end{bmatrix} = \begin{bmatrix} f^{(1)}\big(\mathbf{x}_k^{(1)}, \mathbf{y}_t, \mathbf{s}_k\big) \\ f^{(2)}\big(\mathbf{x}_k^{(2)}, \mathbf{y}_t, \mathbf{s}_k, \mathbf{x}_{k+1}^{(1)}\big) \end{bmatrix}$$

$$\hat{\mathbf{y}}_t = \boldsymbol{C}_{\mathbf{y}}^{(1)}(\mathbf{x}_t^{(1)}) + \boldsymbol{C}_{\mathbf{y}}^{(2)}(\mathbf{x}_t^{(2)}) \tag{3}$$

$$\hat{\lambda}_k = \boldsymbol{C}_{\mathbf{s}}^{(1)}(\mathbf{x}_k^{(1)}) \times \boldsymbol{C}_{\mathbf{s}}^{(2)}(\mathbf{x}_k^{(2)})$$

$$\hat{\mathbf{z}}_k = \boldsymbol{C}_{\mathbf{z}}(\mathbf{x}_k^{(1)})$$

where $f^{(1)}(\cdot)$ and $f^{(2)}(\cdot)$ denote the RNNs that describe the dynamics of the behaviorally relevant neural latents and neural-specific latents, respectively. Note that as the latent state at time $k+1$ is estimated purely using past neural data (that is, $y_1, ..., y_t, s_1, ...s_k$), $\mathbf{x}_{k+1}$ supports causal predictions and is generated by the two RNNs as follows (using Eq. 2),

$$
\begin{aligned}
\mathbf{x}_{k+1}^{(1)} &= f^{(1)}\big(\mathbf{x}_k^{(1)}, \mathbf{y}_t, \mathbf{s}_k\big) = A^{(1)}(\mathbf{x}_k^{(1)}) + \mathbf{Enc_y}^{(1)}(\mathbf{y}_t) + \mathbf{Enc_s}^{(1)}(\mathbf{s}_k) \\
\mathbf{x}_{k+1}^{(2)} &= f^{(2)}\big(\mathbf{x}_k^{(2)}, \mathbf{y}_t, \mathbf{s}_k, \mathbf{x}_{k+1}^{(1)}\big) = A^{(2)}(\mathbf{x}_k^{(2)}) + \mathbf{Enc_y}^{(2)}(\mathbf{y}_t, \mathbf{x}_{t+1}^{(1)}) + \mathbf{Enc_s}^{(2)}(\mathbf{s}_k, \mathbf{x}_{k+1}^{(1)})
\end{aligned}
\tag{4}
$$

In this disentangled form, each RNN, i.e., $f^{(1)}(\cdot)$ and $f^{(2)}(\cdot)$, is parameterized by a recursion function, $A(\cdot)$, and by modality-specific encoders, $\mathbf{Enc_s}(\cdot)$ and $\mathbf{Enc_y}(\cdot)$, for behaviorally relevant neural latents and neural-specific latents, which are denoted by the appropriate superscripts. Further, $C_\mathbf{y}(\cdot)$, $C_\mathbf{s}(\cdot)$, and $C_\mathbf{z}(\cdot)$ are modality-specific decoders that predict the observed multimodal neural and behavior time-series from the two sets of latents. In Eq. 3, $\hat{\lambda}_k$ is the predicted firing rate for Poisson neural modality $\mathbf{s}_k$, $\hat{\mathbf{y}}_t$ is the predicted Gaussian neural modality, and $\hat{\mathbf{z}}_k$ represents the predicted behavior. Note that behavior is predicted from the disentangled behaviorally relevant neural latents while the neural modalities are predicted using both behaviorally relevant neural latents and neural-specific latents; their contributions are fused by adding the Gaussian components and multiplying the Poisson firing-rates (equivalently summing log-rates; see Appendix A.1.2). Also, in our model, we pass $\mathbf{x}_{k+1}^{(1)}$ to the second RNN when computing $\mathbf{x}_{k+1}^{(2)}$ such that it learns the residual neural dynamics not already learned by the first RNN as detailed below and in Appendix A.1.2.

## 3.2 DISENTANGLEMENT OF BEHAVIORALLY RELEVANT NEURAL DYNAMICS

To disentangle behaviorally relevant multimodal neural dynamics from neural-specific dynamics, we learn our model in Eq. 3 with a multi-stage learning approach in three stages. Specifically, we use distinct stages to learn the different subtypes of latent dynamics (i.e., $\mathbf{x}^{(1)}$ and $\mathbf{x}^{(2)}$) such that these subtypes can be disentangled as described below (see Appendix A.1.2 for details). Each stage optimizes a subset of model parameters and subsequent stages hold the learned parameters from previous stages fixed. To make the learning procedure easier to follow, we summarize the overall flow before describing each stage in detail. In the first stage, the model learns a set of neural latents that are predictive of behavior (i.e. behaviorally relevant neural latents). Next, these behaviorally relevant neural latents are used to explain the portion of multimodal neural dynamics that is behavior-related. Finally, the second set of neural latents is learned to model the remaining neural dynamics not explainable by the behaviorally relevant neural latents in the first stage.

**Stage 1 (Supervised training of behaviorally relevant neural latents):** We first train the RNN, $f^{(1)}(\cdot)$ – composed of networks $A^{(1)}(\cdot)$, $\mathbf{Enc_y}^{(1)}(\cdot)$, and $\mathbf{Enc_s}^{(1)}(\cdot)$ – jointly with the behavior decoder $C_\mathbf{z}(\cdot)$. We construct these networks as multi-layer perceptrons (MLP) initialized randomly and optimized jointly to predict the behavior time-series. Specifically, Stage 1 minimizes the negative log-likelihood (NLL) of behavior given the behaviorally relevant neural latents $\mathbf{x}_k^{(1)}$,

$$
\mathbf{L_z} = \sum_k \mathbf{NLL}\big(\mathbf{z}_k; C_\mathbf{z}(\mathbf{x}_k^{(1)})\big)
\tag{5}
$$

This optimization ensures that the learned latent states in this stage capture behaviorally-relevant information in the multimodal neural data. After Stage 1 is complete, the parameters of $f^{(1)}(\cdot)$ and $C_\mathbf{z}(\cdot)$ are fixed.

**Stage 2 (Training neural decoders for behaviorally-relevant latents):** Next, we train the neural decoders, $C_\mathbf{y}^{(1)}(\cdot)$ and $C_\mathbf{s}^{(1)}(\cdot)$, which are again formed as MLPs and predict the neural modalities from the behaviorally relevant neural latents $\mathbf{x}_k^{(1)}$. In this stage, $f^{(1)}(\cdot)$ (and thus $\mathbf{x}_k^{(1)}$) is fixed, and we optimize $C_\mathbf{y}^{(1)}(\cdot)$ and $C_\mathbf{s}^{(1)}(\cdot)$ by minimizing the combined NLL of both modalities using their different likelihood distributions, with a scaling factor $\alpha$ to balance their scale differences to ensure both neural modalities contribute appropriately to the optimization process (Appendix A.1.3).

$$
\mathbf{L_{y,s}}^{(1)} = \sum_t \mathbf{NLL_y}\big(\mathbf{y}_t; C_\mathbf{y}^{(1)}(\mathbf{x}_t^{(1)})\big) + \alpha \sum_k \mathbf{NLL_s}\big(\mathbf{s}_k; C_\mathbf{s}^{(1)}(\mathbf{x}_k^{(1)})\big)
\tag{6}
$$

**Stage 3 (Unsupervised training of neural-specific latents):** We now train the second RNN, $f^{(2)}(\cdot)$ – composed of $A^{(2)}(\cdot)$, $\mathbf{Enc_y}^{(2)}(\cdot)$ and $\mathbf{Enc_s}^{(2)}(\cdot)$ – jointly with the decoders $C_\mathbf{y}^{(2)}(\cdot)$ and $C_\mathbf{s}^{(2)}(\cdot)$

Table 1: **BREM-NET outperforms baselines in neural-behavioral predictions.** Performance of BREM-NET vs. baselines in NHP grid reaching dataset. For all models, 20 spiking channels and 20 LFP channels were used. Mean ± SEM is across 4 sessions and 5 cross-validation folds (N = 20).

| Models | Behavior Decoding (CC) | Neural prediction | |
|---|---|---|---|
| | | LFP (CC) | Spike (PP) |
| MVAE | $0.5040 \pm 0.0095$ | - | - |
| MSID | $0.5442 \pm 0.0161$ | $0.7940 \pm 0.0341$ | $0.3292 \pm 0.0187$ |
| CEBRA | $0.5761 \pm 0.0380$ | $0.4225 \pm 0.0570$ | $0.3113 \pm 0.0164$ |
| MMGPVAE (unsupervised) | $0.5705 \pm 0.0415$ | $0.7477 \pm 0.0777$ | $0.2867 \pm 0.0144$ |
| MMGPVAE (supervised) | $0.6201 \pm 0.0437$ | $0.7454 \pm 0.0776$ | $0.2716 \pm 0.0148$ |
| mmPLRNN (unsupervised) | $0.5329 \pm 0.0524$ | $0.7329 \pm 0.0803$ | $0.2982 \pm 0.0184$ |
| mmPLRNN (supervised) | $0.6804 \pm 0.0417$ | $0.5972 \pm 0.1007$ | $0.2692 \pm 0.0184$ |
| **BREM-NET** | $\mathbf{0.7645 \pm 0.0052}$ | $\mathbf{0.8078 \pm 0.0129}$ | $\mathbf{0.3746 \pm 0.0036}$ |

by minimizing the combined NLL of both neural modalities given both behaviorally relevant neural latents ($\mathbf{x}_k^{(1)}$) and neural-specific latents($\mathbf{x}_k^{(2)}$). Since $C_{\mathbf{y}}^{(1)}(\cdot)$, and $C_{\mathbf{s}}^{(1)}(\cdot)$ and the value of $\mathbf{x}_k^{(1)}$ are already learned in Stages 1 and 2 and account for the part of $\mathbf{y}_t$ and $\mathbf{s}_k$ that are already predictable with behaviorally relevant neural latents, this stage is trained to explain any residual dynamics in the neural data that $\mathbf{x}_k^{(1)}$ does not explain. Thus, to learn these residual dynamics, we form the loss as:

$$\mathbf{L}_{\mathbf{y},\mathbf{s}}^{(2)} = \sum_t \mathbf{NLL}_{\mathbf{y}}\big(\mathbf{y}_t; C_{\mathbf{y}}^{(1)}(\mathbf{x}_t^{(1)}), C_{\mathbf{y}}^{(2)}(\mathbf{x}_t^{(2)})\big) + \alpha \sum_k \mathbf{NLL}_{\mathbf{s}}\big(\mathbf{s}_k; C_{\mathbf{s}}^{(1)}(\mathbf{x}_k^{(1)}), C_{\mathbf{s}}^{(2)}(\mathbf{x}_k^{(2)})\big) \quad (7)$$

Overall, the explicit disentanglement is achieved by using the above three-stage process, with each stage performed sequentially until convergence. This process takes in three modalities – spikes, LFPs, and behavior – and learns their evolution with a unified nonlinear dynamical model in disentangled form. Stage 1 fuses information across neural modalities to learn the multimodal behaviorally relevant neural latents that predict behavior. Then, Stage 2 learns how to predict the multimodal neural time-series from these behaviorally-relevant neural latents. Finally, Stage 3 learns the multimodal neural-specific latents by predicting the variability in multimodal neural time-series that is not already predictable from the behaviorally relevant neural latents learned in Stage 1 (see Table A.4 for each stage's contribution).

# 4 EXPERIMENTAL RESULTS

We evaluate BREM-NET on two distinct nonhuman primate (NHP) datasets from different laboratories and involving different behavioral tasks, neural signal modalities, and experimental paradigms, as well as on simulated data (see Appendix A.5.2 for simulation results). We compare BREM-NET against several recent multimodal baselines, and perform ablation studies to gain insight into BREM-NET's improved performance and quantify the impact of design components.

## 4.1 MULTIMODAL NEURAL FUSION IN THE NHP GRID REACHING TASK

We first applied BREM-NET to a publicly available dataset of an NHP performing a 2D grid-reaching task in virtual reality (O'Doherty et al., 2020). This dataset contains simultaneous recordings of discrete spiking activity and continuous LFPs from the primary motor cortex (M1) while the animal performed sequential 2D reaching movements to random targets (Fig. 2a, Appendix A.3.1). We treated spike counts and raw LFPs as neural time-series and used the 2D cursor velocity in the $x$ and $y$ directions as behavior signals, which were decoded from latents inferred by BREM-NET through fusion of neural modalities. For all sessions, we fixed the total latent dimensionality at 64 ($n_x = 64$), allocating 16 dimensions for behaviorally relevant latents ($n_1 = 16$). These fixed choices were made based on initial exploration of the model on a single session of the dataset (Fig. A.5a-d, Appendix A.1.4) and were kept consistent across all models, including baselines and ablations.

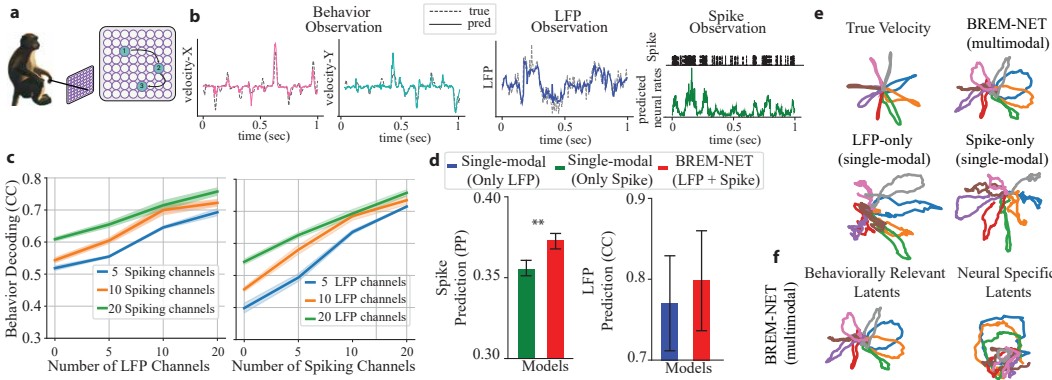

Figure 2: **BREM-NET successfully fuses multimodal neural time-series, thus improving neural-behavioral predictions and disentangled latent state extraction in the NHP grid reaching dataset.** (a) Dataset and task visualization. (b) True behavior, LFP, and spike time-series (dashed) versus their predictions with BREM-NET (solid) for a representative session corresponding to the results in Tab. 1. (c) *left*: Cross-validated behavior decoding CC when 5, 10, or 20 spiking channels were the primary modality and an increasing number of LFP channels were fused with them (N=20). Lines represent the mean and shaded areas show the SEM. *right*: Similar to *left* when LFP channels were the primary modality. (d) *left*: Cross-validated Predicted Power (PP) of spikes using BREM-NET vs. when training a single-modal model on the spiking modality alone. *right*: Cross-validated Prediction CC of LFP modality with BREM-NET vs. when training a single-modal model with the LFP modality alone (N=20). Bars represent the mean and error bars represent the SEM. Asterisks indicate significance of comparison (**: $p < 0.001$ one-sided Wilcoxon signed-rank test) (e) Behaviorally relevant latents inferred by BREM-NET are more congruent with true velocity than behaviorally relevant latents infered from single-modal models (LFP-only or Spike-only). (f) True velocity trajectories are much more congruent with behaviorally relevant latent trajectories than with neural-specific latent trajectories, confirming disentanglement of latent states by BREM-NET.

### 4.1.1 MULTIMODAL FUSION ENHANCES THE ACCURACY OF DISENTANGLED LATENTS AND BEHAVIOR DECODING

To assess multimodal neural fusion for learning behaviorally relevant neural dynamics and neural-specific dynamics, we conducted a series of experiments where either spikes or LFPs served as the primary modality, and increasing numbers of channels from the other modality were fused (Fig. 2c). When spiking activity served as the primary modality, the addition of LFP channels consistently enhanced behavior decoding across varying spike channel counts (Fig. 2c, left, $p < 10^{-5}$, $n = 20$, one-sided Wilcoxon signed-rank test). A similar trend was observed when LFPs were used as the primary modality (Fig. 2c, right, $p < 10^{-5}$, $n = 20$, one-sided Wilcoxon signed-rank test). Notably, the improvements were more pronounced when LFP channels were the primary modality, suggesting that spiking activity encoded more behavior-related information compared to LFPs in this dataset. In addition to using 20 spiking and 20 LFP channels as done for the main analyses, we also performed an extended analysis using all available channels in the dataset for completeness and generalization to higher-dimensional datasets. In this scenario, we again observed that BREM-NET resulted in improvements over single-modal variants, while this improvement was somewhat smaller as expected in a high-information regime for each modality (Appendix A.5.4).

We further examined the effect of multimodal neural fusion on the quality of latent representations by comparing the latent trajectories produced by single-modal models with those from BREM-NET (Fig. 2e, details for latent trajectory plots provided in Appendix A.5.6). As shown in Fig. 2e, the disentangled behaviorally relevant latent trajectories $\mathbf{x}^{(1)}$ inferred via multimodal spike-LFP fusion by BREM-NET were more congruent with behavior trajectories (velocity) than those inferred from either the spike or LFP modality alone. Also, consistent with decoding results, latents inferred using only spikes were more aligned with behavior trajectories than those inferred using only LFPs (Fig. 2e). Combined with the more accurate multimodal behavior decoding results in Fig. 2c, these

latent visualizations show that multimodal fusion enhances the interpretability and relevance of the disentangled latent trajectories in the context of behavior. As further evidence of disentanglement, we evaluated the behavior prediction of the learned behaviorally relevant neural latents($\mathbf{x}^{(1)}$) and neural-specific latents ($\mathbf{x}^{(2)}$) separately. As shown in Fig. A.3b, $\mathbf{x}^{(1)}$ accurately decodes behavior, while $\mathbf{x}^{(2)}$ excels in predicting neural activity yet fails to decode behavior accurately (0.7629 vs. 0.3367 decoding correlation coefficient (CC) with $\mathbf{x}^{(1)}$ and $\mathbf{x}^{(2)}$, respectively ($p < 10^{-7}$, $n = 20$, one-sided Wilcoxon signed-rank test). These results indicate that BREM-NET achieves disentanglement of behaviorally relevant neural dynamics in $\mathbf{x}^{(1)}$ during Stage 1, while $\mathbf{x}^{(2)}$ captures neural-specific dynamics that help in neural prediction. Furthermore, the latent trajectories in Fig. 2f show that true velocity is more congruent with behaviorally relevant neural latents ($\mathbf{x}^{(1)}$) than neural-specific latents ($\mathbf{x}^{(2)}$), further supporting the disentangled structure of the latent space learned by BREM-NET.

### 4.1.2 BREM-NET IS ROBUST TO ASYNCHRONOUS SAMPLING RATES AND DROPOUTS

We evaluated the robustness of BREM-NET when neural modalities are partially dropped out or sampled at different rates (different temporal resolutions). First, in the NHP grid-reaching dataset, we downsampled LFPs by a factor of 5. As shown in Tab. A.6, BREM-NET maintained its behavior decoding performance in this asynchronous setting ($p > 0.5$, $n = 20$, two-sided Wilcoxon signed-rank test), confirming that it does not require the modalities to be strictly aligned in time. Next, we introduced stochastic modality-specific dropouts by randomly dropping spike or LFP samples at each timestep with probability $p \in \{0.2, 0.4, 0.6, 0.8\}$. This setup simulates realistic challenges such as modality-specific noise, artifacts, or recording failures. As demonstrated in Tab. A.7, BREM-NET showed strong robustness. Even for spikes that have a primary role in behavior decoding (Fig. 2e), performance dropped by less than 5% even with 40% spike dropout, and further our method was more robust to LFP dropouts. Overall, these results show that BREM-NET can robustly fuse even asynchronous and incomplete neural modalities.

### 4.1.3 BREM-NET OUTPERFORMS BASELINES AND ABLATED VARIANTS

Next, we conducted a comprehensive ablation study to quantify the contribution of architectural components in BREM-NET (Tab. A.4). First, BREM-NET outperformed its fully linear counterpart (L-BREM-NET). Second, BREM-NET surpassed single-modal variants that used only LFP or spikes without any fusion, confirming the benefit of multimodal neural fusion while disentangling behaviorally relevant neural dynamics. We also evaluated an unsupervised variant (U-BREM-NET) in which Stages 1 and 2 were omitted. While U-BREM-NET matched BREM-NET on neural predictions, BREM-NET consistently outperformed U-BREM-NET in behavioral decoding ($p < 10^{-5}$, $n = 20$, one-sided Wilcoxon signed-rank test), showing the importance of disentanglement by training the dynamical model simultaneously with different neural modalities along with behavior. In addition, we evaluated another variant of BREM-NET in which Stage 3 was excluded (BREM-NET w/o Stage 3). While this variant matched BREM-NET in behavior decoding, its neural prediction accuracy was substantially lower ($p < 10^{-7}$, $n = 20$, one-sided Wilcoxon signed-rank test), showing the importance of learning neural-specific dynamics as well as behaviorally relevant ones for accurate prediction of both neural-behavioral data.

To assess the consistency of disentangled, behaviorally-relevant neural latent representations across sessions, we also extended BREM-NET to a multi-session setting. Specifically, we introduced session-specific linear projection heads to account for inter-session variability, while keeping the core model architecture shared across sessions. Our hypothesis was that if a unified model—with the same latent dimensionality as the single-session variant—could accurately decode behavior across sessions, it would indicate that a consistent latent structure is recoverable across sessions. To test this, we pooled data from multiple sessions and trained a model with the same number of latent states used in single-session experiments. Despite

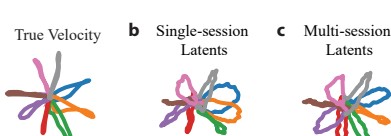

Figure 3: **Multi-session fitting extracts consistent latents across session.** Visualization of behaviorally-relevant latent trajectories under single-session and multi-session fitting. Latents from the multi-session model are smoother and again closely track the true velocity.

increased data heterogeneity, the multi-session model achieved comparable behavior decoding accuracy (0.7648 vs. 0.7645 CC), and its latent trajectories were congruent with the true behavior and

smoother than those from the single-session model (Fig. 3). This result suggests the consistency of behaviorally-relevant neural latent representations across sessions and the ability of our model to recover these consist latents.

Finally, we compared BREM-NET's neural-behavioral prediction performance against several recent baselines (Tab. 1). First, we compared BREM-NET to MMGPVAE (Gondur et al., 2023) and mmPLRNN (Kramer et al., 2021), both of which utilize two modalities during training and inference (see Related Work). For a fair comparison, we used the two neural modalities (spike and LFP) as input signals for both baselines and trained two variations of them: 1) unsupervised training, which is the current version of these models, 2) supervised training, where we extended these models by adding the behavior reconstruction loss to their original loss (see Appendix A.2.8, A.2.9 for details). BREM-NET more accurately predicted behavior and neural activity (Tab. 1), regardless of whether we trained MMGPVAE or mmPLRNN supervised or unsupervised. We also compared to MSID (Ahmadipour et al., 2024), a linear multimodal dynamical model that integrates two neural modalities but lacks supervision and disentanglement. BREM-NET achieved superior neural-behavioral predictions (Tab. 1). In addition, we compared BREM-NET to CEBRA (Schneider et al., 2023), which uses contrastive loss on behavior to extract embeddings (see Appendix A.2.7). We used the multisession training of CEBRA to train it jointly using LFP and spikes and found that BREM-NET outperforms CEBRA in both behavior and neural prediction. Finally, we compared with MVAE, a generic variational autoencoder trained without temporal modeling or behavioral supervision that can account for multimodal datasets (Wu & Goodman, 2018). MVAE showed the weakest performance, underscoring the importance of dynamical modeling of neural-behavioral data. Overall, BREM-NET significantly outperformed all baselines for behavior decoding and neural prediction ($p < 0.0001$, $n = 20$, one-sided Wilcoxon signed-rank test). Beyond predictions, Fig. A.4 shows that latents learned by BREM-NET are more congruent with behavior compared to baselines, confirming BREM-NET's success in learning behaviorally relevant representations.

Together, these results show that combining supervised disentanglement and multimodal nonlinear neural fusion in BREM-NET are important for learning latents and predicting neural-behavioral data.

### 4.2 MULTIMODAL NEURAL FUSION IN A DISTINCT NHP CENTER-OUT REACHING TASK

To evaluate the generalizability of BREM-NET, we next applied it to another distinct NHP dataset (Flint et al., 2012), which differs in both behavioral paradigm and recording setup. In contrast to the previous dataset, this dataset involves an NHP performing center-out reaches to randomly presented peripheral targets using a manipulandum interface. Multimodal neural recordings included discrete spiking activity and LFP power bands from both M1 and premotor (PMd) cortices, thus providing distinct neural signals compared to the previous dataset that included spiking and raw LFP signals (not power features). We used the 2D manipulandum velocity in the $x$ and $y$ directions as our behavior signals (Fig. A.1a, see Appendix A.3.2 for details). We trained BREM-NET with $n_1 = 16$ and $n_x = 32$ (Fig. A.5e-g). Similar to the NHP grid reaching dataset, behavior decoding performance improved through multimodal neural fusion (Fig. A.1c, $p < 10^{-5}$, $n = 15$, one-sided Wilcoxon signed-rank test). Further, again our method disentangled behaviorally relevant neural dynamics as confirmed in Fig. A.3d. Finally, in this dataset, BREM-NET's behavior decoding again outperformed the baselines (Tab. A.3, $p < 0.0001$, $n = 15$, one-sided Wilcoxon signed-rank test). While some unsupervised or contrastive baselines outperformed BREM-NET in neural prediction in this dataset, they achieved much lower behavior decoding performance than BREM-NET.

## 5 DISCUSSION

We introduced BREM-NET, a nonlinear dynamical model that integrates multiple neural modalities such as LFPs and spikes alongside behavioral data within a unified framework. We built our model as an RNN with disentangled behaviorally relevant and neural-specific states, and incorporated modality-specific encoders and decoders to account for distinct characteristics in multimodal neural time-series and achieve multimodal neural fusion. Across two independent multimodal NHP neural datasets, we showed that our method not only disentangles behaviorally relevant dynamics in multimodal neural data, but also fuses behaviorally relevant information across neural modalities, leading to better latent state, neural, and behavior inferences. We also demonstrated that BREM-NET can achieve these gains even when modalities exhibit distinct temporal resolutions or have modality-specific sample

dropouts, which are natural challenges in real-world neuroscience applications such as BCIs. We also found that our method can learn consistent disentangled behaviorally relevant representations across sessions by extending it to a multi-session setting using session-specific projections.

There are several future directions to extend our method's utility. First, we selected the dimensionality of our shared latent space based on behavior decoding performance in a validation set in a single session of the datasets. Incorporating statistical estimators such as in (Giaffar et al., 2024) can provide a principled alternative to our current performance-based selection for determining the shared latent dimensionality. Furthermore, we applied our method to spikes, raw LFPs, or LFP power feature modalities given their importance in neuroscience and BCIs. Exploring our method's extension across more modalities, such as EEG, intracranial EEG, or even physiological modalities such as heart rate is important in future direction. Also, future work can extend BREM-NET to study shared latent dynamics across multiple brains, such as in social interaction paradigms. In such settings, a disentangled multimodal model could help isolate task-relevant dynamics shared across brains from those private to an individual brain. As another exciting future direction, BREM-NET can be extended to multi-region modeling by introducing a shared global latent $x^{(1)}$ that captures cross-region dynamics, while assigning region-specific latents $x^{(2)}$ for the dynamics that are not shared across regions. The shared latents would represent population-level coordination—analogous to a communication subspace—while region-specific latents would capture local dynamics unique to each area.

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

## A    APPENDIX

### A.1    METHOD DETAILS

#### A.1.1    METHOD FORMULATION

In Equation 3, we provided a compact formulation of our RNN-based architecture. Here, we present a more detailed formulation that explicitly describes the disentangled latent dynamics and the associated model components used in BREM-NET. The full model is defined as:

$$
\begin{cases}
\begin{bmatrix} \mathbf{x}_{k+1}^{(1)} \\ \mathbf{x}_{k+1}^{(2)} \end{bmatrix} = \begin{bmatrix} A^{(1)}(\mathbf{x}_k^{(1)}) \\ A^{(2)}(\mathbf{x}_k^{(2)}) \end{bmatrix} + \begin{bmatrix} \mathbf{Enc}_{\mathbf{y}}^{(1)}(\mathbf{y}_t) + \mathbf{Enc}_{\mathbf{s}}^{(1)}(\mathbf{s}_k) \\ \mathbf{Enc}_{\mathbf{y}}^{(2)}(\mathbf{y}_t, \mathbf{x}_{t+1}^{(1)}) + \mathbf{Enc}_{\mathbf{s}}^{(2)}(\mathbf{s}_k, \mathbf{x}_{k+1}^{(1)}) \end{bmatrix} \\
\hat{\mathbf{y}}_t = \boldsymbol{C}_{\mathbf{y}}^{(1)}(\mathbf{x}_t^{(1)}) + \boldsymbol{C}_{\mathbf{y}}^{(2)}(\mathbf{x}_t^{(2)}) \\
\hat{\lambda}_k = \boldsymbol{C}_{\mathbf{s}}^{(1)}(\mathbf{x}_k^{(1)}) \times \boldsymbol{C}_{\mathbf{s}}^{(2)}(\mathbf{x}_k^{(2)}) \\
\hat{\mathbf{s}}_k \sim \mathcal{P}ois(\hat{\lambda}_k) \\
\hat{\mathbf{z}}_k = \boldsymbol{C}_{\mathbf{z}}(\mathbf{x}_k^{(1)})
\end{cases}
\tag{8}
$$

where $A^{(1)}(\cdot)$, and $A^{(2)}(\cdot)$ represents the state transition function, capturing the temporal evolution of the latent state, and $\mathbf{Enc}_{\mathbf{y}}^{(1)}(\cdot)$, $\mathbf{Enc}_{\mathbf{s}}^{(1)}(\cdot)$, $\mathbf{Enc}_{\mathbf{y}}^{(2)}(\cdot)$, and $\mathbf{Enc}_{\mathbf{s}}^{(2)}(\cdot)$ are modality-specific encoders for $\mathbf{y}_t$ and $\mathbf{s}_k$, respectively. We use distinct time indices $t$ and $k$ for $y$ and $s$ respectively to indicate that neural modalities can be sampled at different temporal resolutions.

Here, the overall latent state $\mathbf{x}_k \in \mathbb{R}^{n_x}$, which captures the underlying dynamics of the neural-behavioral data, is designed to disentangle behaviorally relevant dynamics, represented by $\mathbf{x}_k^{(1)} \in \mathbb{R}^{n_1}$, from multimodal neural-specific dynamics, represented by $\mathbf{x}_k^{(2)} \in \mathbb{R}^{n_x - n_1}$. We also note that each $\mathbf{x}_k$ is estimated purely using past neural data (that is, $y_1, ..., y_{t-1}, s_1, ... s_{k-1}$), so $\mathbf{x}_{k+1}$ has the information about neural modalities up to time $k$. As such, our model supports causal predictions.

Lines 2-4 of Equation 8 describe how the model predicts observations:

**2**: $\hat{\mathbf{y}}_t$ is the predicted Gaussian neural signal, combining contributions from both latent states $\mathbf{x}_k^{(1)}$ and $\mathbf{x}_k^{(2)}$ through the nonlinear decoders $\boldsymbol{C}_{\mathbf{y}}^{(1)}(\cdot)$ and $\boldsymbol{C}_{\mathbf{y}}^{(2)}(\cdot)$. This prediction is performed in Stage 2 and Stage 3 of the architecture (Figure 1).

**3**: $\hat{\lambda}_k$ represents the predicted firing rate of spike counts, which also combines $\mathbf{x}_k^{(1)}$ and $\mathbf{x}_k^{(2)}$ through the nonlinear decoders $\boldsymbol{C}_{\mathbf{s}}^{(1)}(\cdot)$ and $\boldsymbol{C}_{\mathbf{s}}^{(2)}(\cdot)$. Notably, unlike the Gaussian decoders $\boldsymbol{C}_{\mathbf{y}}^{(1)}(\cdot)$ and $\boldsymbol{C}_{\mathbf{y}}^{(2)}(\cdot)$, our Poisson decoders $\boldsymbol{C}_{\mathbf{s}}^{(1)}(\cdot)$ and $\boldsymbol{C}_{\mathbf{s}}^{(2)}(\cdot)$, employ an $\exp(.)$ nonlinearity output layer activation function to force non-negativity of Poisson rates. Therefore, Poisson decoders can be expressed as:

$$
\boldsymbol{C}_{\mathbf{s}}^{(1)}(\cdot) = \exp\left(\boldsymbol{C}_{\mathbf{s}}''^{(1)}(\cdot)\right)
$$
$$
\boldsymbol{C}_{\mathbf{s}}^{(2)}(\cdot) = \exp\left(\boldsymbol{C}_{\mathbf{s}}''^{(2)}(\cdot)\right)
\tag{9}
$$

where, $\boldsymbol{C}_{\mathbf{s}}''(\cdot)$ has no output layer activation function. As a result, predicted firing rate can be formulated as:

$$
\hat{\lambda}_k = \exp\left(\boldsymbol{C}_{\mathbf{s}}''^{(1)}(\mathbf{x}_k^{(1)}) + \boldsymbol{C}_{\mathbf{s}}''^{(2)}(\mathbf{x}_k^{(2)})\right)
\tag{10}
$$

This step is also done in Stage 2 and Stage 3 of the architecture (Figure 1).

**4**: $\hat{\mathbf{z}}_k$ is the predicted behavior, decoded from the low-dimensional latents of Stage 1 using the nonlinear decoder $C_{\mathbf{z}}(\cdot)$ learned during Stage 1 (Figure 1).

### A.1.2  LEARNING STAGES

This section details the three-stage optimization procedure used to train BREM-NET as described in Equation 8. Each stage is designed to optimize distinct components of the model in order to accurately predict observations while disentangling behaviorally relevant and neural-specific dynamics. Across all stages, the negative log-likelihood (NLL) of the observed data is used as the training objective.

**Stage 1:**

First, we fit an RNN, composed of $A^{(1)}(\cdot)$ and modaltiy-specific encoders, $\mathbf{Enc}_{\mathbf{y}}^{(1)}(\cdot)$ and $\mathbf{Enc}_{\mathbf{s}}^{(1)}(\cdot)$, jointly with the behavior decoder network $C_{\mathbf{z}}(\cdot)$, using $n_1$ states, to minimize behavior prediction NLL given past neural activities $\mathbf{y}_k$, and $\mathbf{s}_k$ (Equation 11). This ensures that the neural dynamics that are predictive of behavior are learned. The resulting latent states, $\mathbf{x}_k^{(1)}$, form the first set of latents in BREM-NET which are behaviorally relevant latents. The optimization can be expressed as:

$$\begin{cases} \mathbf{x}_{k+1}^{(1)} = A^{(1)}(\mathbf{x}_k) + \mathbf{Enc}_{\mathbf{y}}^{(1)}(\mathbf{y}_t) + \mathbf{Enc}_{\mathbf{s}}^{(1)}(\mathbf{s}_k) \\ \hat{\mathbf{z}}_k = C_{\mathbf{z}}(\mathbf{x}_{\mathbf{k}}^{(\mathbf{1})}) \\ \mathbf{L}_z^{(1)} = -\sum_k \log\left( p_z(\mathbf{z}_k \mid \mathbf{x}_k^{(1)})\right) \end{cases} \tag{11}$$

where $p_z$ represent the conditional likelihood of behavior $\mathbf{z}_k$ given behaviorally relevant latents in Stage 1.

**Stage 2:**

Next, we fit two modality-specific decoders to the latents $\mathbf{x}_k^{(1)}$ to predict the two neural modalities. The objective is to simultaneously minimize NLL of both neural modalities, ensuring accurate predictions across these diverse data types. Since BREM-NET accommodates neural modalities from different statistical distributions, the loss function must reflect these differences. To address this, we optimize a weighted combination of the NLLs for the two modalities, incorporating a scaling factor $\alpha$ to balance the disparity in scales between them (Eq. 12). This $\alpha$ factor ensures that both modalities contribute meaningfully to the optimization process, regardless of their inherent scale.

$$\begin{cases} \mathbf{x}_{k+1}^{(1)} = A^{(1)}(\mathbf{x}_k) + \mathbf{Enc}_{\mathbf{y}}^{(1)}(\mathbf{y}_t) + \mathbf{Enc}_{\mathbf{s}}^{(1)}(\mathbf{s}_k) \\ \hat{\bar{\mathbf{y}}}_t = C_{\mathbf{y}}^{(1)}(\mathbf{x}_t^{(1)}) \\ \hat{\bar{\lambda}}_k = C_{\mathbf{s}}^{(1)}(\mathbf{x}_k^{(1)}) \\ \mathbf{L}_{\mathbf{y},\mathbf{s}}^{(1)} = -\left( \sum_t \log\left( p_y(\mathbf{y}_t \mid \mathbf{x}_t^{(1)})\right) + \alpha \sum_k \log\left( p_s(\mathbf{s}_k \mid \mathbf{x}_k^{(1)})\right)\right) \end{cases} \tag{12}$$

Here, $p_y$ and $p_s$ represent the conditional likelihoods of the two neural modalities, $\mathbf{y}_k$ and $\mathbf{s}_k$, respectively. This stage ensures that $\mathbf{x}_k^{(1)}$ adequately explains the behaviorally relevant components of the neural data. Therefore, $\hat{\bar{\mathbf{y}}}_k$, and $\hat{\bar{\lambda}}_k$ correspond to the part of neural modalities $\mathbf{y}_k$, and $\mathbf{s}_k$, that are predictable with behaviorally relevant latents.

**Stage 3:**

In the final stage, we learn the second RNN, composed of $A^{(2)}(\cdot)$ and modaltiy-specific encoders, $\mathbf{Enc}_{\mathbf{y}}^{(2)}(\cdot)$ and $\mathbf{Enc}_{\mathbf{s}}^{(2)}(\cdot)$ jointly with the modality-specific neural decoders $C_{\mathbf{y}}^{(2)}(\cdot)$ and $C_{\mathbf{s}}^{(2)}(\cdot)$ to predict the neural-specific dynamics of the two neural modalities that are independent of behavior. We represent the unpredicted part of multimodal neural activities with $\tilde{\mathbf{y}}_k, \tilde{\mathbf{s}}_k$, such that the full neural predictions are given by:

$$\hat{\mathbf{y}}_t = \hat{\bar{\mathbf{y}}}_t + \hat{\tilde{\mathbf{y}}}_t$$
$$\hat{\mathbf{s}}_k \sim \mathcal{P}ois(\hat{\bar{\lambda}}_k \times \hat{\tilde{\lambda}}_k) \tag{13}$$

This final stage ensures that the neural-specific dynamics (not explained by behavior) are also captured using separate set of latents. The optimization for this stage is defined as:

$$
\begin{cases}
\mathbf{x}_{k+1}^{(2)} = A^{(2)}(\mathbf{x}_k) + \mathbf{Enc}_{\mathbf{y}}^{(2)}(\mathbf{y}_t) + \mathbf{Enc}_{\mathbf{s}}^{(2)}(\mathbf{s}_k) \\
\quad \hat{\bar{\mathbf{y}}}_t = \boldsymbol{C}_{\mathbf{y}}^{(2)}(\mathbf{x}_t^{(2)}) \\
\quad \hat{\bar{\lambda}}_k = \boldsymbol{C}_{\mathbf{s}}^{(2)}(\mathbf{x}_k^{(2)}) \\
\mathbf{L}_{\mathbf{y},\mathbf{s}}^{(2)} = -\big(\sum_t \log\big(p_y(\mathbf{y}_t \mid \mathbf{x}_t^{(1)}, \mathbf{x}_t^{(2)})\big)\big) + \alpha \sum_k \log\big(p_s(\mathbf{s}_k \mid \mathbf{x}_k^{(1)}, \mathbf{x}_k^{(2)})\big)\big)
\end{cases}
\tag{14}
$$

### A.1.3 SCALING FACTOR $\alpha$

To address the differences in scale between the likelihoods of two neural modalities, we use a scaling factor, $\alpha$, in the loss function to ensure that both neural modalities contribute proportionally to the optimization process. To determine the value of $\alpha$, we first compute the average value across all available time steps for each neural modality. This average serves as a baseline, representing the overall scale or magnitude of the observed data for that modality. Then, assuming the mean value as the predicted value for each modality, we calculate the log-likelihood of the observed data. This step quantifies the "scale" of each modality in terms of their likelihood under their respective distributions. Finally, the scaling factor $\alpha$ is determined as the ratio of the likelihoods of the modalities. Specifically, $\alpha$ is set such that it normalizes the contributions of the modalities, effectively equalizing their influence during the optimization process.

### A.1.4 HYPERPARAMETERS AND IMPLEMENTATION

For BREM-NET, we employed consistent neural network architectures across both simulated and real-world evaluations. The specific hyperparameters employed for training on the simulation dataset, as well as analyses of both real-world datasets, are detailed in Table A.1. For encoders and decoders in our model i.e., $\mathbf{Enc}_{\mathbf{s}}^{(1)}(\cdot)$, $\mathbf{Enc}_{\mathbf{y}}^{(2)}(\cdot)$, $\mathbf{Enc}_{\mathbf{s}}^{(2)}(\cdot)$, $\boldsymbol{C}_{\mathbf{y}}^{(1)}(\cdot)$, $\boldsymbol{C}_{\mathbf{s}}^{(1)}(\cdot)$, $\boldsymbol{C}_{\mathbf{y}}^{(2)}(\cdot)$, $\boldsymbol{C}_{\mathbf{s}}^{(2)}(\cdot)$ we used multi-layer perceptrons (MLPs) with each having specific number of hidden layers and neurons (Table A.1). Otherwise, for our L-BREM-NET analysis, we remove the hidden layers and nonlinearity in feedforward neural networks. Notably, unlike the Gaussian decoders $\boldsymbol{C}_{\mathbf{y}}^{(1)}(\cdot)$ and $\boldsymbol{C}_{\mathbf{y}}^{(2)}(\cdot)$, our Poisson decoders $\boldsymbol{C}_{\mathbf{s}}^{(1)}(\cdot)$ and $\boldsymbol{C}_{\mathbf{s}}^{(2)}(\cdot)$ employ an $\exp(.)$ nonlinearity output layer activation function to force non-negativity of Poisson rates. We adopted a Step Decay learning rate scheduler Ge et al. (2019), which is known for its effectiveness in training deep neural networks. It was initialized with a learning rate of $1e-3$, which was halved ($\gamma = 0.5$) every 400 steps. To ensure stable training and convergence, we initialized all model parameters using the Xavier-normal initialization method citeglorot2010understanding. For optimization, we utilized the AdamW optimizer Loshchilov & Hutter (2017). We trained model for a maximum of 1000 epochs to ensure thorough learning, while employing early stopping to avoid overfitting. All models were trained on CPU servers equipped with AMD EPYC 7513 and 7542 processors (2.90 GHz, 32 cores) with parallelization. Our implementation is available at `https://anonymous.4open.science/r/BREM-NET-OP18`.

In simulation experiments, the state dimensions were aligned with those of the true underlying model. For real-world data analyses, we explored the impact of varying the state dimensions $n_1$ and $n_x$ in one session of data. Specifically, we tested dimensions $n_1, n_x \in [2, 4, 8, 16, 32, 64, 128]$ and report the corresponding results in Figure A.5 for one dataset. After choosing the best $n_1$ and $n_x$ based on one session of data, we kept it consistent for all models and baselines.

Table A.1: **Hyperparameter settings of BREM-NET.** Encoders and decoders architectures, as well as training configurations, used across all experiments and models.

| Hyperparameter | Value |
|---|---|
| Encoders ($\mathbf{Enc_y}, \mathbf{Enc_s}$) | [64, 128] |
| Neural decoders ($C_\mathbf{y}, C_\mathbf{s}$) | [128, 128, 128] |
| Behavior decoder ($C_z$) | [64] |
| Initial learning rate | $1e-3$ |
| LR scheduler | StepLR |
| Batch size | 32 |
| Number of epochs | 1000 |
| Weight decay | $1e-3$ |

### A.1.5 MODEL COMPLEXITY ANALYSIS

In this section, we report the number of trainable parameters for each model. These counts are provided to document the complexity of BREM-NET compared to our baselines(Table A.2).

Table A.2: **Parameter counts for models.** Number of trainable parameters for each model.

| Model | Number of parameters |
|---|---|
| MMGPVAE | 103,578 |
| mmPLRNN | 353,328 |
| CEBRA | 697,344 |
| BREM-NET | 218,946 |

### A.1.6 EVALUATION METRICS

After learning BREM-NET, we utilize the learned nonlinear modality-specific encoders $\mathbf{Enc_y^{(1)}}(\cdot)$, $\mathbf{Enc_s^{(1)}}(\cdot)$, $\mathbf{Enc_y^{(2)}}(\cdot)$, $\mathbf{Enc_s^{(2)}}(\cdot)$, and recursion networks $A^{(1)}(\cdot)$, $A^{(2)}(\cdot)$ to infer 1-step-ahead predicted latent states using Equation 3 for the held-out test data in cross-validation. Using these inferred latent states, we compute the predicted neural activity and behavior by applying the corresponding decoders $C_\mathbf{y}(\cdot)$, $C_\mathbf{s}(\cdot)$, and $C_\mathbf{z}(\cdot)$. For behavior predictions, we refer to this process as *decoding*, as the model exclusively utilizes neural modalities, $\mathbf{Y}$ and $\mathbf{S}$, and does not incorporate behavior itself as an input during inference.

To evaluate model performance, we use 5-fold cross-validation for both simulation and real data analyses. For LFP neural activity and continuous behavioral signals, which are modeled using a Gaussian distribution, we report the Pearson Correlation Coefficient (CC) in all of our analysis (Tables 1, A.4, A.3) between predicted and actual observations.

For spike count data, modeled with a Poisson distribution, we quantify one-step-ahead prediction accuracy using the Prediction Power (PP) metric, defined as $\mathbf{PP = 2\,AUC - 1}$, where $\mathbf{AUC}$ is the area under the curve of the receiver operating characteristic (ROC) curve Macke et al. (2011). The ROC curve is constructed by using the one-step-ahead predicted firing rates ($\hat{\lambda}_{k+1}$) as classification scores to determine whether a time step contains a spike Truccolo et al. (2010).

## A.2 BASELINES

To evaluate the performance of BREM-NET and highlight its contributions, we compare it against a diverse set multimodal baseline models. For a fair comparison, all models and baselines are trained with the same latent dimensionality and evaluated using the same cross-validation protocol and number of channels per modality.

First, to assess the role of nonlinear modeling in BREM-NET, we compare it against two fully linear dynamical methods: MSID Ahmadipour et al. (2024) and a fully linear version of BREM-NET, denoted L-BREM-NET, in which all nonlinear components are removed. Second, to highlight the

significance of using multimodal neural modalities on the neural-behavioral dynamics, we include single-modal variants of our model trained with either LFP or spike data alone. These baselines help highlight the benefit of combining neural modalities when modeling behaviorally relevant dynamics. Third, to evaluate the importance of behavior supervision and disentanglement, we include an unsupervised variant of our model, denoted U-BREM-NET, in which behavior is not used during training. In addition, we include a variant, denoted BREM-NET w/o Stage 3, in which Stage 3 is omitted. We further compare BREM-NET against several recent works developed for neural-behavioral modeling, including: mmPLRNN Kramer et al. (2021), MMGPVAE Gondur et al. (2023), and CEBRA Schneider et al. (2023). Finally, we include MVAE Wu & Goodman (2018), a generic multimodal variational autoencoder baseline that aligns latent spaces across modalities without modeling dynamics and not developed for neural-behavioral modeling.

### A.2.1   L-BREM-NET

L-BREM-NET serves as a simplified linear baseline. It preserves the same multi-stage architecture and overall structure of BREM-NET but replaces all encoders and decoders with linear transformations. This baseline uses identical loss functions and optimization procedures to ensure fair comparison. Nonlinear activations and hidden layers are removed, making the model purely linear while maintaining the core structure of our approach.

### A.2.2   U-BREM-NET

U-BREM-NET is an unsupervised baseline that performs only the third stage of the BREM-NET learning procedure (see Appendix A.1.2). Unlike BREM-NET, U-BREM-NET learns the neural dynamics without any supervision or consideration of their relevance to behavior. Behavior information is not utilized during the learning process, and the extracted latent states are subsequently mapped to behavior data through a downstream decoder. Effectively, U-BREM-NET represents a special case of BREM-NET where $n_1 = 0$, meaning that no behaviorally relevant latent states are disentangled during training.

### A.2.3   BREM-NET W/O STAGE 3

This ablated variant of BREM-NET includes the first two stages of BREM-NET where Stage 3 is omitted. Specifically, the model learns the behavior-relevant latent states $x^{(1)}$ with $n_1 = 16$ and behavior decoder $C_z$ from multimodal neural input in stage 1, and fits neural decoders ($C_y$ and $C_s$) in stage 2 which is used for prediction of multimodal neural activities. The neural-specific latent states $x^{(2)}$, introduced in Stage 3 of the full model, is removed in this variant. Consequently, the model lacks a dedicated mechanism for modeling components of neural activity that are independent of behavior.

### A.2.4   SINGLE-MODAL MODELS

For single-modal models, we adopt a simplified version of the architecture shown in Figure 1, where the encoder and decoder components corresponding to the additional neural modality are removed. In this configuration, the model focuses solely on one modality of neural data, such as LFPs or spike counts and learns behaviorally relevant dynamics from that single modality and thus does not fuse multimodal neural during inference. The first stage of the model is dedicated to modeling the latent structure of the chosen modality, capturing the most informative features that are behaviorally relevant. Subsequently, these learned representations are used to predict the associated behavioral signals, under the assumption that the chosen modality alone contains sufficient information to account for the observed behavior. This single-modality setup serves as a baseline comparison to our multimodal model, enabling an evaluation of the individual contribution of each neural modality. By isolating the dynamics of a single modality, we are able to assess its relative effectiveness in capturing behavior-related information and gain insight into the strengths and limitations of using single neural modality for behavior decoding.

### A.2.5 MSID

Multiscale Subspace Identification (MSID) is a linear model designed for analyzing multiscale dynamical systems in neural activity, operating under the assumption of linear dynamics Ahmadi et al. (2021). In our analysis, we compared the performance of BREM-NET against MSID and demonstrated that, unlike the linear approach of MSID, the nonlinear information aggregation and behavior supervision property of BREM-NET significantly enhances both behavior decoding and neural prediction. We used the official implementation of MSID provided by the authors for training [1]. For training MSID, we set the horizon hyperparameters as specified in the original manuscript Ahmadipour et al. (2024), specifically $h_y = h_z = 10$.

### A.2.6 MVAE

Multimodal Variational Autoencoder (MVAE) Wu & Goodman (2018) is a VAE-based architecture designed to model multimodal data using a mixture-of-experts formulation over the posterior distributions. MVAE is trained to align latent representations across modalities. To adapt MVAE for our use case, we treated each time-step as an independent data point and trained the model in a non-sequential manner, effectively bypassing the lack of a dynamical modeling backbone. Behavior decoding was performed by fitting linear regression models on the inferred latent representations. Since MVAE does not model temporal dependencies, it is not capable of performing neural time-series prediction, which requires explicit dynamical modeling.

### A.2.7 CEBRA

CEBRA Schneider et al. (2023) is a recent method designed to extract latent embeddings from neural recordings, optionally guided by behavioral information. It employs a 1D convolutional neural network to extract embeddings from small windows of neural data, and optimized through a contrastive objective. In the supervised variant (CEBRA-Behavior), behavior labels are used to define positive and negative sample pairs, encouraging the model to learn embeddings that are discriminative with respect to behavior and thus behaviorally relevant. In the unsupervised version, positive samples are determined based on time-difference to the anchor sequence.

Unlike RNN-based methods, CEBRA models the temporal dynamics of neural activity through 1D convolutional layers. Further, CEBRA architecture do not include decoder networks but it only consists of encoder networks to extract latent states from input signals. To enable performance comparisons on behavior decoding and neural prediction tasks, we therefore fit separate linear regression models on top of the CEBRA embeddings to decode neural and behavior signals. For a fair comparison, we use the multisession version of CEBRA that jointly processes both continuous LFP and discrete spike data, which generates consistent embeddings across the two recording modalities. This allows CEBRA to operate as a multimodal model in our experiments. We follow the original implementation and adopt the default hyperparameters provided in the original work for CEBRA[2].

### A.2.8 MMPLRNN

Multi-modal piecewise-linear RNN (mmPLRNN) is a variational method previously introduced for multimodal dynamical modeling with piecewise-linear RNNs Kramer et al. (2021). mmPLRNN builds on a prior work, PLRNN Durstewitz (2017), by fusing information from two modalities. By design, mmPLRNN utilizes two modalities during inference. To enable a fair comparison with BREM-NET, which does not incorporate behavior during inference, we trained two versions of mmPLRNN. In the first version, mmPLRNN was trained in an unsupervised manner, using only neural modalities (e.g., LFP and spiking signals) as input during training and inference. For the second version, we extended mmPLRNN to a supervised framework. Specifically, we introduced an additional behavior prediction objective during training. This was achieved by learning a feedforward neural network that maps the inferred latent factors from mmPLRNN to the behavior signals. The behavior reconstruction loss was added to the mmPLRNN's training objective to encourage the latent space to capture behaviorally relevant dynamics. Importantly, the behavior signal was not treated as an input to the model. Instead, the goal was to infer latent factors from neural modalities (LFP and

---

[1]We use the implementation provided in https://github.com/ShanechiLab/MultiscaleSID

[2]We use the implementation provided in https://github.com/AdaptiveMotorControlLab/CEBRA

spikes) that could accurately reconstruct the behavior. However, unlike BREM-NET, this extended mmPLRNN approach does not disentangle the latent space into behaviorally relevant and behaviorally irrelevant components. Instead, it only encourages the learned latent space to be more behaviorally informative overall, without explicit disentanglement.

Furthermore, mmPLRNN does not support predicted inference of latent factors. Thus, to provide a fair comparison between all baseline methods, we modified mmPLRNN's inference algorithm to infer one-step-ahead predicted latent factors. For each timestep $k$, we passed the neural modalities up to timestep $k - 1$ to the inference network, and forward propagated the latent states at $k - 1$ through PLRNN dynamics to obtain one-step-ahead predicted latent states for $k$. After this procedure is applied for all time horizon, we then obtained one-step-ahead predicted neural modalities by passing the predicted latent factors through decoder networks. Further, the original mmPLRNN implementation supports only Gaussian and categorical observation models. To accommodate our experimental setup, we implemented a Poisson observation model following the methodology outlined in Appendix C of Kramer et al. (2021). Furthermore, we trained mmPLRNN models with nonlinear readouts comparable to BREM-NET and evaluated their neural-behavioral prediction performance. We used the official implementation and recommended hyperparameters of mmPLRNN for the aforementioned extensions and data analyses[3].

### A.2.9  MMGPVAE

The Multimodal Gaussian Process Variational Autoencoder (MMGPVAE) is a recent multimodal framework that leverages Gaussian processes to model the latent distribution underlying multimodal observations Gondur et al. (2023). MMGPVAE's inference network first extracts the frequency content of the latent factors and subsequently converts them into time-domain representations, rather than directly estimating them in the time domain. This approach allows MMGPVAE to filter out high-frequency components from the latent factors, resulting in smoother representations. As noted in the authors' manuscript, the choice of a Gaussian Process (GP) prior is particularly advantageous in experimental settings where the latent dynamics are assumed to be smooth Gondur et al. (2023).

Similar to mmPLRNN, MMGPVAE utilizes two modalities during inference. Thus, for the comparisons shown in Tables 1, A.3, we followed the same procedure as for MMPLRNN-unsupervised and supervised variations. To ensure consistency with BREM-NET, we trained MMGPVAE with 64-dimensional latent factors per modality, where 32 out of the 64 dimensions were shared across modalities. All MMGPVAE variations were trained for 100 epochs. We also modified the encoder/decoder architecture, as the default configuration led to suboptimal performance on our dataset. The new configuration is the same as what we use in BREM-NET (refer to Table A.1). Similar to mmPLRNN, MMGPVAE also does not support predicted inference of latent states. To infer one-step-ahead predicted latent states, for each timestep $k$, we replaced the neural modalities after timestep $k - 1$ with zeros such that the latent states at $k$ is inferred only by neural observations up to timestep $k - 1$ (i.e., one-step-ahead predicted). Then, we passed these neural signals to MMGPVAE's inference network, and we obtained one-step-ahead predicted latent states. After this procedure is applied for all time horizon, we then obtained one-step-ahead predicted neural modalities by passing the predicted latent states through decoder networks. Furthermore, we extended MMGPVAE to a supervised setting similar to the supervised extension of mmPLRNN, see Appendix A.2.8 for details. We used the implementation of MMGPVAE provided by the authors for our analysis [4].

### A.3  ADDITIONAL INFORMATION ON REAL DATASETS

### A.3.1  NONHUMAN PRIMATE (NHP) GRID REACHING DATASET

We analyzed a publicly available dataset O'Doherty et al. (2020) in which a macaque monkey (Monkey I), performed a grid reaching task. Neural activity was recorded from the primary motor cortex (M1) using a 96-channel electrode array as the subject controlled a 2D cursor to reach randomly appearing targets on a grid within a virtual reality environment. Targets were presented sequentially, with no intervening time gaps between their appearances. In our analysis, took the subject's 2D fingertip velocity recorded at a 10 ms timescale as the behavior time-series to decode.

---

[3]We use the implementation provided in https://github.com/DurstewitzLab/mmPLRNN

[4]We use the implementation provided in https://github.com/RabiaGondur/MM-GPVAE

For the spiking data, we utilized multi-unit spiking activity recorded at a 10 ms timescale. From the four recording sessions, we selected the top 20 spiking channels based on their behavior prediction accuracies. For the local field potential (LFP) data, signals were extracted from the raw neural recordings using a low-pass filter with a 300 Hz cut-off frequency. The filtered signals were downsampled to 100 Hz (10 ms timescale). Similar to the spiking data, we identified the top 20 LFP channels across the four sessions based on their behavior prediction accuracies. We report the mean and SEM computed across four sessions and five cross-validated folds (N=20).

### A.3.2 NHP CENTER-OUT REACHING DATASET

In this publicly available dataset Flint et al. (2012), a macaque monkey (Monkey C) performed a 2D center-out reaching task while grasping a two-link manipulandum. The task involved reaching from a central position to one of eight outer targets arranged in a circle, followed by returning to the center to begin the next trial. In our analysis, we used the 2D manipulandum velocity recorded at a 10 ms timescale as the behavior variable to decode. Neural recordings were obtained from a 96-channel silicon microelectrode array (Blackrock Microsystems) chronically implanted in the primary motor (M1) and premotor (PMd) cortices of the monkey's arm.

For the spiking data, we utilized multi-unit spiking activity recorded at a 10 ms timescale. From the three recording sessions, we selected the top 20 spiking channels based on their behavior prediction accuracies. For the local field potential (LFP) data, raw signals were band-pass filtered between 0.5 and 500 Hz and sampled at 2 kHz. LFP power was computed across five frequency bands: 0-4 Hz, 7-20 Hz, 70-115 Hz, 130-200 Hz, and 200-300 Hz, using a 256 ms window with 10 ms resolution, yielding signals at a 100 Hz. Similar to the spiking data, we identified the top 20 LFP channels across the three sessions based on their behavior prediction accuracies. We report the mean and SEM computed across three sessions and five cross-validated folds (N=15).

### A.4 SIMULATION DETAILS

For our simulation data in section A.5.2, we generated Gaussian and Poisson observations from four randomly initialized systems as defined by the dynamics in

$$\begin{aligned}
\mathbf{x}_{k+1} &= F(\mathbf{x}_k) + \mathbf{w}_k \\
\mathbf{y}_k \mid \mathbf{x}_k &\sim \mathcal{N}\big(\mathbf{y}_k \mid \boldsymbol{C}_\mathbf{y}(\mathbf{x}_k)\big) \\
\mathbf{s}_k \mid \mathbf{x}_k &\sim \mathcal{P}ois\big(\mathbf{s}_k \mid \boldsymbol{C}_\mathbf{s}(\mathbf{x}_k)\big) \\
\mathbf{z}_k \mid \mathbf{x}_k &\sim \mathcal{N}\big(\mathbf{z}_k \mid \boldsymbol{C}_\mathbf{z}(\mathbf{x}_k)\big)
\end{aligned} \tag{15}$$

In this setup, the latent state evolves in time according to a recursion function $F(\cdot)$ with additive noise $\mathbf{w}_k$. $\mathbf{y}_k$ represents Gaussian neural activity, $\mathbf{s}_k$ corresponds to spiking neural activity, and $\mathbf{z}_k$ denotes behavior. A total of eight latent variables ($n_x = 8$) were drawn from the system, where $n_1 = 4$ latent dimensions were allocated to the behaviorally relevant subspace and the remaining $n_2 = n_x - n_1 = 4$ to the behaviorally independent subspace. These latent variables were then mapped to their corresponding observations using nonlinear functions $\boldsymbol{C}_\mathbf{y}(\cdot)$, $\boldsymbol{C}_\mathbf{s}(\cdot)$, and $\boldsymbol{C}_\mathbf{z}(\cdot)$. For each randomly initialized system, we generated a total of $2e6$ samples. These simulation settings demonstrate the benefits of fusing multimodal information from two neural modalities while disentangling the behaviorally relevant subspace, leading to improved predictions of both neural and behavior time-series.

## A.5 Supplementary Results

### A.5.1 NHP Center-out Reaching Task Results

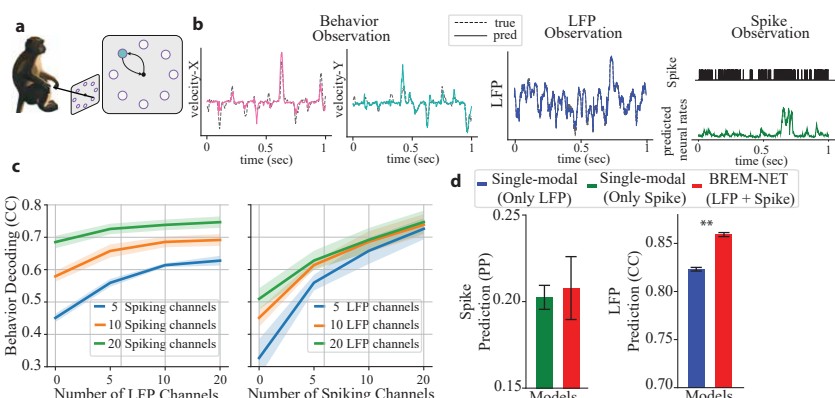

Figure A.1: **BREM-NET again improves neural-behavioral predictions due to multimodal neural fusion in another NHP dataset with a center-out reaching task.** **(a)** Dataset and task visualization. **(b)** True behavior, LFP, and spike time-series (dashed) versus their predictions with BREM-NET (solid) for a representative session corresponding to the results in Table 1. **(c)** *left*: Cross-validated behavior decoding CC when 5, 10, or 20 spiking channels were the primary modality and an increasing number of LFP channels were fused with them (N=15). Lines represent the mean and shaded areas show the SEM. *right*: Similar to *left* when LFP channels were the primary modality. **(d)** *left*: Cross-validated Prediction CC of LFP modality with BREM-NET vs. when training a single-modal model with the LFP modality alone (N=15). Bars represent the mean and error bars represent the SEM. *right*: Cross-validated Predicted Power (PP) of firing rates using BREM-NET vs. when training a single-modal model on the spiking modality alone. Asterisks indicate significance of comparison (**: $p < 0.001$, one-sided Wilcoxon signed-rank test)

Table A.3: **Comparison of BREM-NET to baselines in neural-behavioral predictions.** Prediction performance of BREM-NET compared to baselines in NHP center-out reaching dataset. For all multimodal models, 20 spiking channels as well as 20 LFP channels were used. For single-modal baselines, 20 channels of the chosen modality were used. Mean ± SEM is across 3 sessions and 5 cross-validation folds (N = 15).

| Models | Behavior Decoding (CC) | Neural prediction | |
| --- | --- | --- | --- |
| | | LFP (CC) | Spike (PP) |
| L-BREM-NET | $0.6559 \pm 0.0227$ | $0.8118 \pm 0.0126$ | $0.0521 \pm 0.0068$ |
| U-BREM-NET | $0.6481 \pm 0.0268$ | $0.8558 \pm 0.0044$ | $0.2077 \pm 0.0106$ |
| Single-modal (LFP) | $0.5250 \pm 0.0329$ | $0.8220 \pm 0.0021$ | - |
| Single-modal (Spike) | $0.6823 \pm 0.0192$ | - | $0.1879 \pm 0.0212$ |
| MSID | $0.5583 \pm 0.0821$ | $0.8228 \pm 0.0116$ | $0.1595 \pm 0.0363$ |
| CEBRA | $0.5880 \pm 0.0367$ | $0.3991 \pm 0.0435$ | $\mathbf{0.3160 \pm 0.0367}$ |
| MMGPVAE (unsupervised) | $0.6094 \pm 0.0872$ | $\mathbf{0.8751 \pm 0.0135}$ | $0.2793 \pm 0.0748$ |
| MMGPVAE (supervised) | $0.6455 \pm 0.0869$ | $0.7772 \pm 0.0235$ | $0.2001 \pm 0.0350$ |
| mmPLRNN (unsupervised) | $0.5912 \pm 0.0276$ | $0.8217 \pm 0.0132$ | $0.2232 \pm 0.0259$ |
| mmPLRNN (supervised) | $0.7362 \pm 0.0563$ | $0.5868 \pm 0.0219$ | $0.0580 \pm 0.0420$ |
| **BREM-NET** | $\mathbf{0.7554 \pm 0.0230}$ | $0.8366 \pm 0.0074$ | $0.2002 \pm 0.0083$ |

### A.5.2 OUR METHOD FUSES MULTIMODAL DATA, THUS BETTER PREDICTING ALL MODALITIES IN SIMULATIONS

We first validated the ability of BREM-NET to effectively fuse information across multiple modalities through a series of simulation experiments. These simulations involved three modalities with shared and independent dynamics. The underlying dynamical system used to generate the synthetic time-series for these modalities is defined in Equation 15. We set the total simulated neural latent dimensionality to $n_x = 8$, with four dimensions allocated to behaviorally relevant latents ($n_1 = 4$) and the remaining four to neural-specific latents ($n_2 = 4$).

We generated observations from both Gaussian and Poisson distributions using 100 channels each and compared BREM-NET to single-modal models that only used either Gaussian or Poisson observations alongside behavior (Appendix A.2.4). Specifically, in Figure A.2a, we chose a primary neural modality, Poisson in the left panel and Gaussian on the right panel. We then fused this primary modality with gradually increasing numbers of channels of the other secondary modality, as shown on the x axis. As such, the points corresponding to 0 on the x axis indicate the single-modal performance and all other points show BREM-NET that fuses different numbers of channels of the two different modalities.

We found that BREM-NET demonstrated superior performance in behavior decoding compared to single-modal models, as shown in Figure A.2a. In scenarios where the Poisson modality served as the primary modality, incorporating Gaussian channels consistently improved behavior decoding accuracy (Figure A.2a, left). Similar trend was observed when the Gaussian modality was the primary modality and Poisson channels were fused (Figure A.2a, right). Further, incorporating information from one neural modality enhanced the prediction accuracy of the other modality, showing the power of cross-modality information fusion (Figure A.2b). Consistently, BREM-NET successfully predicted both neural time-series modalities and behavioral time-series from the learned latent states (Figure A.2c).

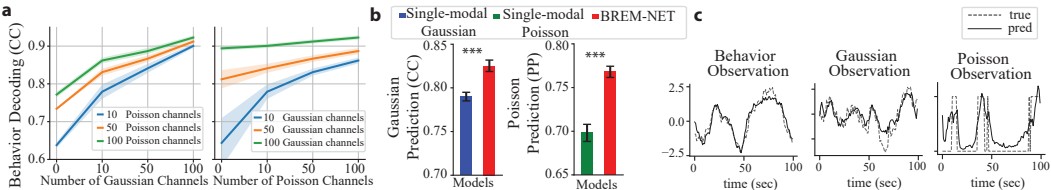

Figure A.2: **BREM-NET improves behavior decoding and multimodal neural predictions in a simulated dataset.** Cross-validated accuracy of behavior decoding and Gaussian and Poisson neural prediction in four simulated systems (N=20). **(a)** *left*: behavior decoding correlation coefficient (CC) when 10, 50, or 100 Poisson channels were the primary modality and an increasing number of Gaussian channels were fused with them. Lines represent the mean and shaded areas show the standard error of the mean (SEM). *right*: Similar to *left* when Gaussian channels were the primary modality. **(b)** *left*: Prediction CC of Gaussian modality using BREM-NET vs. when training a single-modal model with Gaussian modality alone. Bars represent the mean and error bars show the SEM. *right*: Predicted Power (PP) of the Poisson modality using BREM-NET vs. when training a single-modal model on the Poisson modality alone. Asterisks indicate significance of comparison (***: $p < 10^{-7}$ one-sided Wilcoxon signed-rank test)**(c)** True versus predicted time-series using BREM-NET, showing the model's ability to accurately predict all observation modalities.

### A.5.3 Ablation studies

Table A.4: **Ablations of BREM-NET.** Prediction performance of BREM-NET compared to ablated versions in NHP grid reaching dataset. For all multimodal models, 20 spiking channels as well as 20 LFP channels were used. For single-modal baselines, 20 channels of the chosen modality were used. Mean ± SEM is across 4 sessions and 5 cross-validation folds (N = 20). **L-BREM-NET**: A linear version of BREM-NET. **U-BREM-NET**: An unsupervised version trained without behavior supervision or disentanglement (i.e., BREM-NET w/o Stage 1,2). **BREM-NET w/o Stage 3**: Trained stage 1 and 2 without learning neural-specific latents in Stage 3. **Single-modal (LFP/Spike)**: Variants of BREM-NET using only one neural modality with behavior. **BREM-NET**: Full three-stage architecture with multimodal fusion, disentanglement, and modality-specific modeling. BREM-NET achieves the highest performance across all metrics, validating the importance of each design choice.

| Models | Behavior Decoding (CC) | Neural prediction | |
| --- | --- | --- | --- |
| | | LFP (CC) | Spike (PP) |
| L-BREM-NET | $0.6547 \pm 0.0341$ | $0.7998 \pm 0.0623$ | $0.1060 \pm 0.0380$ |
| U-BREM-NET | $0.7098 \pm 0.0087$ | $\mathbf{0.8410 \pm 0.0104}$ | $0.3791 \pm 0.0053$ |
| BREM-NET w/o Stage 3 | $0.7580 \pm 0.0088$ | $0.5571 \pm 0.0111$ | $0.3087 \pm 0.0068$ |
| Single-modal (LFP) | $0.5424 \pm 0.0346$ | $0.7702 \pm 0.0587$ | - |
| Single-modal (Spike) | $0.6078 \pm 0.0185$ | - | $0.3485 \pm 0.0142$ |
| **BREM-NET** | $\mathbf{0.7584 \pm 0.0085}$ | $\mathbf{0.8350 \pm 0.0117}$ | $\mathbf{0.3796 \pm 0.0058}$ |

### A.5.4 Effect of using all recording channels

We performed an additional analysis using all 96 spiking and 96 LFP channels and obtained a behavior decoding (CC) of $0.7711 \pm 0.0177$, slightly higher than $0.7645 \pm 0.0052$ with the top 20 channels (Fig. 2c). The small improvement indicates that including low-SNR or noisy channels does not significantly enhance decoding performance, as these channels contain limited behaviorally relevant information. This justifies our choice to focus on the top predictive channels for improved efficiency with minimal performance trade-off. Nevertheless, BREM-NET still has benefits over single-modal variants without any preselection and using all channels as shown below:

Table A.5: **BREM-NET still has benefits over single-modal variants without any preselection and using all channels.** Prediction performance of BREM-NET compared to single-modal variants in NHP grid reaching dataset. For BREM-NET, all 96 spiking channels as well as 96 LFP channels were used. For single-modal variants, 96 channels of the chosen modality were used. Mean ± SEM is across 4 sessions and 5 cross-validation folds (N = 20).

| Models | Behavior Decoding (CC) |
| --- | --- |
| Single-modal (LFP) | $0.5752 \pm 0.0325$ |
| Single-modal (Spike) | $0.7575 \pm 0.0189$ |
| BREM-NET (multimodal) | $\mathbf{0.7711 \pm 0.0179}$ |

### A.5.5 Robustness to missing data, temporal resolution mismatch, and dropout

In practical neural recording scenarios, different modalities—such as LFPs and spikes—are often sampled at different temporal resolutions due to hardware limitations or preprocessing constraints. Additionally, missing data are common in real-world datasets due to noise, sensor dropout, or transmission errors. Although the datasets used in our experiments are fully observed, BREM-NET is explicitly designed to handle missing data and asynchronous sampling as explained in Methods section 3. This is achieved through modality-specific encoder networks, which flexibly incorporate available modalities at each time step. If a modality is missing, its encoder output can be set to zero, allowing the latent dynamics to update using only the available inputs.

To empirically test robustness to mismatched sampling rates, we downsampled the LFP signals by a factor of 5 in the NHP grid-reaching dataset, while leaving the spike and behavior data unchanged. This setup mimics real-world conditions where continuous signals are collected at lower frequencies than discrete spike trains, or where LFP observations may be intermittently unavailable. As shown in

Table A.6, behavior decoding and spike prediction performance remained largely stable, whereas LFP prediction degraded due to masked predictions on downsampled timesteps. This confirm that BREM-NET is resilient to both temporal resolution mismatches and partial observability, effectively integrating modalities sampled at different rates without requiring strict alignment.

In addition to resolution mismatches, we also evaluated robustness under irregular and stochastic missing data. At each timestep, either spikes or LFPs were randomly dropped with probability $p \in \{0.2, 0.4, 0.6, 0.8\}$, creating asynchronous gaps where only one modality was available. Unlike the resolution mismatch experiment, where missing samples occur regularly, this dropout analysis simulates challenges such as channel noise, transient artifacts, or partial recording failures that occur irregularly. As shown in Table A.7, decoding performance remained remarkably stable across dropout probabilities, particularly for LFPs. Even when 80% of LFP samples were removed, behavior decoding declined by less than 2%. Spike dropouts produced a larger impact, consistent with spikes carrying richer behavior-related information in this dataset (Fig. 2e). Still, the performance degradation remained modest, with less than 5% drop at 40% spike dropout. Together, these analyses demonstrate that BREM-NET flexibly integrates multimodal inputs under asynchronous sampling and partial observability, maintaining robust decoding accuracy under realistic and adverse recording conditions.

Table A.6: **BREM-NET is robust to temporal resolution mismatch and missing observations.** We compare BREM-NET trained under two conditions: (1) **same-resolution**, where both LFP and spike data are aligned and sampled at the same rate, and (2) **different-resolution**, where LFP signals are downsampled by a factor of 5 to simulate asynchronous sampling. These results demonstrate that BREM-NET can effectively integrate multimodal signals under realistic constraints such as sampling mismatch and missing observations without compromising decoding performance.

| Models | Behavior Decoding (CC) | Neural prediction | |
| --- | --- | --- | --- |
| | | LFP (CC) | Spike (PP) |
| BREM-NET w/ different-resolution | $0.7571 \pm 0.0051$ | $0.6272 \pm 0.0234$ | $0.3701 \pm 0.0031$ |
| BREM-NET w/ same-resolution | $0.7645 \pm 0.0052$ | $0.8078 \pm 0.0125$ | $0.3746 \pm 0.0036$ |

Table A.7: **BREM-NET is robust to dropouts.** We evaluated decoding performance when either spikes or LFPs were randomly droped at each timestep with probability $p \in \{0.2, 0.4, 0.6, 0.8\}$. BREM-NET remained highly resilient, particularly to LFP dropouts. Spike dropouts produced a larger effect, consistent with their primary role in behavior decoding These results demonstrate that BREM-NET flexibly integrates asynchronous and incomplete modalities while maintaining robust behavior decoding performance.

| Dropout probability | Behavior Decoding (CC) | |
| --- | --- | --- |
| | LFP dropout | Spike dropout |
| 0.0 | $0.7645 \pm 0.0052$ | $0.7645 \pm 0.0052$ |
| 0.2 | $0.7613 \pm 0.0061$ | $0.7543 \pm 0.0042$ |
| 0.4 | $0.7601 \pm 0.0071$ | $0.7245 \pm 0.0063$ |
| 0.6 | $0.7573 \pm 0.0067$ | $0.6993 \pm 0.0072$ |
| 0.8 | $0.7516 \pm 0.007$ | $0.6292 \pm 0.0065$ |

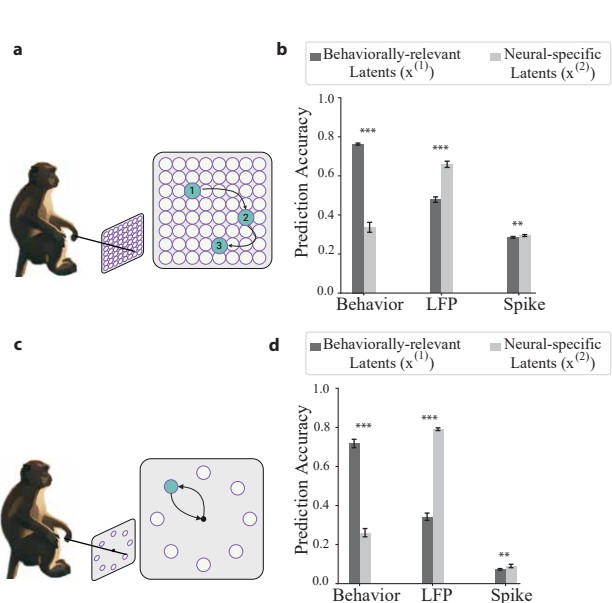

Figure A.3: **Disentanglement of behaviorally relevant and neural-specific latent dynamics through multi-stage learning.** Evaluation of the disentanglement of the learned latent states by using $\mathbf{x}^{(1)}$ and $\mathbf{x}^{(2)}$ separately to predict behavior and neural modalities. **(a)** NHP grid-reaching dataset with $n_1 = 16$, $n_x = 32$. **(b)** Behavior decoding is substantially more accurate when using the behaviorally relevant latents $\mathbf{x}^{(1)}$ compared to the neural-specific latents $\mathbf{x}^{(2)}$ (0.7629 vs. 0.3367). Conversely, neural prediction accuracy improves when using $\mathbf{x}^{(2)}$, demonstrating its specialization in capturing neural-specific information. Together, these results confirm that Stage 1 successfully extracts behaviorally relevant dynamics into $\mathbf{x}^{(1)}$, while Stage 3 captures complementary neural-specific components in $\mathbf{x}^{(2)}$ that enhance neural prediction performance. **(c–d)** same as a-b for NHP center-out reaching dataset with $n_1 = 4$, $n_x = 16$. Asterisks indicate significance of comparison (***: $p < 10^{-7}$, **: $p < 0.001$ one-sided Wilcoxon signed-rank test)

### A.5.6 LATENT TRAJECTORY PLOTTING DETAILS

To visualize the learned latent states, we plotted the condition-averaged latent trajectories in grid reaching dataset O'Doherty et al. (2020). Trials were grouped into 8 discrete categories based on the direction of the reach movement, corresponding to the target directions in the task. For each condition, we computed the average latent trajectory across all trials within that condition. To enable 2D visualization, we applied Principal Component Analysis (PCA) on the full set of inferred $\mathbf{x}_k^{(1)}$ latents across all trials and retained the first two principal components. These top two components were then used to project the condition-averaged latent trajectories into a 2D plane. Each trajectory was plotted using a unique color corresponding to the reach direction condition. For comparison, we also plotted the condition-averaged true behavior trajectories (e.g., 2D hand velocity) using the same procedure, aligning and averaging behavioral traces over trials per direction. This provides an intuitive comparison of whether the latent dynamics learned by the model align with true task structure.

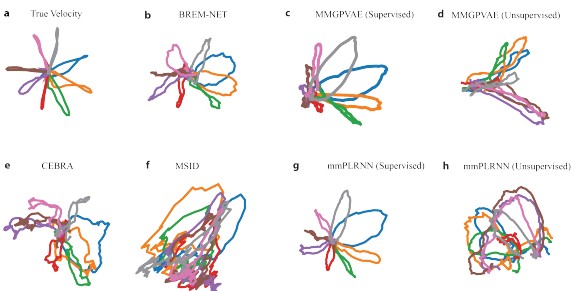

Figure A.4: **Latent state trajectories on NHP reaching dataset.** (a) True condition-averaged reach velocity trajectories. Here reach trials were divided into 8 conditions based on reach direction (color coded). (b) Condition-averaged trajectories of behaviorally relevant latents in BREM-NET. (c) Condition-averaged trajectories in MMGPVAE (Supervised). (d) Condition-averaged trajectories in MMGPVAE (Unsupervised). (e) Condition-averaged trajectories in CEBRA. (f) Condition-averaged trajectories in MSID. (g) Condition-averaged trajectories in mmPLRNN (Supervised). (h) Condition-averaged trajectories in mmPLRNN (Unsupervised).

### A.5.7 LATENT DIMENSIONALITY SELECTION

To select the latent dimensionalities for each real-world dataset, we evaluated model performance on one session from each dataset across a range of dimensionalities. Specifically, we varied the number of behaviorally relevant latent dimensions $n_1$ and the total latent dimensions $n_x$ in $[2, 4, 8, 16, 32, 64, 128]$, and assessed their impact on behavior decoding and neural prediction accuracy. In BREM-NET, Stage 1 is designed to extract behaviorally relevant latent states, which are used exclusively for behavior decoding. Therefore, we selected $n_1$ based on the behavior decoding performance. As shown in Figure A.5b,f, performance peaks at $n_1 = 16$ for both datasets, which we selected as the behaviorally relevant latent dimensionality. Stage 3 is responsible for modeling neural-specific dynamics to improve neural predictions beyond what is captured by behaviorally relevant latents. Thus, we selected $n_x$ based on neural prediction performance. As shown in Figure A.5c,d,g,h, neural prediction performance peaks around $n_x = 64$ for the first dataset and around $n_x = 32$ for the second dataset, which we used as the total latent dimensionalities for subsequent analyses.

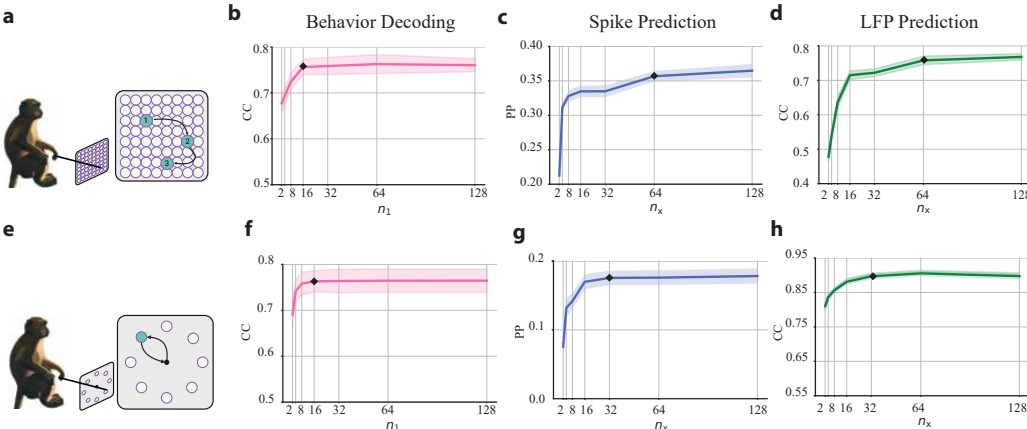

Figure A.5: **Selecting latent dimensions based on behavior decoding and neural prediction performance in one session of datasets.** This figure illustrates how behavior decoding and neural prediction accuracy vary with the number of latent dimensions. **(a)** NHP grid reaching dataset. **(b)** Behavior decoding performance (CC) peaks around $n_1 = 16$, which is therefore selected as the number of behaviorally relevant latent dimensions. **(b, c)** Neural prediction accuracy—measured via Gaussian prediction CC and spike prediction PP—peaks around $n_x = 64$, which is selected as the total latent dimensionality. **(e)** NHP center-out reaching dataset. **(f)** Behavior decoding performance (CC) peaks around $n_1 = 16$, which is therefore selected as the number of behaviorally relevant latent dimensions. **(g, h)** Neural prediction accuracy—measured via Gaussian prediction CC and spike prediction PP—peaks around $n_x = 32$, which is selected as the total latent dimensionality.

### A.5.8 LATENT INFORMATIVENESS FOR BEHAVIOR AND MULTIMODAL NEURAL TIMESERIES

As described in Appendix A.5.7, for the first dataset we fixed the total latent dimensionality to 64, allocating 16 dimensions to behaviorally relevant latents learned in Stage 1. To quantify how much information each of the 64 latent dimensions carries about behavior and neural activity, we conducted a new latent–informativeness analysis. We constructed cumulative sets of latents by adding latents one by one to the sets; for example, set $i$ includes latents $\{1, \ldots, i\}$, with latents $1, \ldots, 16$ being the behaviorally relevant neural latents and latents $17, \ldots, 64$ being the neural-specific ones. We then trained nonlinear MLP decoders on each set of latents and evaluated the behavior decoding performance as well as spike and LFP prediction performances using those latents. Plotting each of these performances versus the number of included latents yielded the cumulative informativeness plots shown in Figure A.6. This figure shows the following:

1. **Behavior information is highly concentrated in latents learned in Stage 1.** The cumulative behavior informativeness curve rises sharply within the first ∼10–16 latent dimensions and then saturates (Figure A.6 (a)). Adding the remaining latents (up to 64 total) provides no additional improvement in behavior decoding, indicating that the behaviorally relevant information in neural data is captured by the behaviorally relevant neural latents learned in Stage 1.

2. **Neural informativeness accumulates gradually across all 64 latents (Stages 1 and 3).** For neural activity, informativeness increases steadily as more neural-specific latents are included (Figure A.6 (b,c)). This demonstrates that the neural-specific latents learned in Stage 3 capture residual neural variability that is distinct from the behaviorally relevant neural latents in Stage 1.

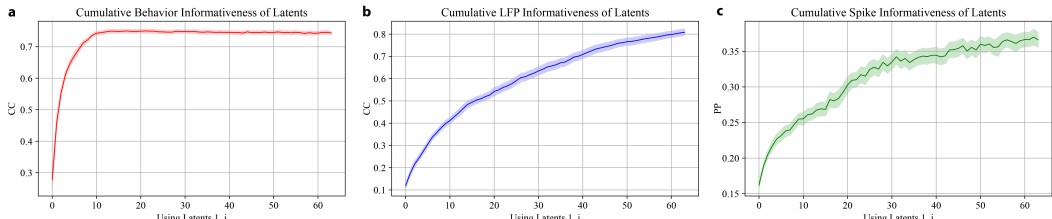

Figure A.6: **Cumulative latent informativeness for behavior, LFP, and spike activity. (a)** Behavior decoding (CC) saturates within the first ∼ 10–16 latent dimensions, indicating that behaviorally relevant information in multimodal neural data is captured in Stage 1 with 16 latent dimensions. **(b,c)** Informativeness for LFP (CC) and spikes (PP) increases gradually across the full 64-dimensional latent space, reflecting that neural-specific latents in Stage 3 provide distinct residual information about multimodal neural dynamics that is not captured in the behaviorally relevant neural latents. Together, these results provide further quantitative evidence for effective disentanglement in BREM-NET.

### A.5.9 ADDITIONAL COMPARISON WITH CEBRA

In CEBRA, the latents are mapped to behavior and neural activity with a linear projection layer. In order to make a fair comparison with this baselines that has linear decoders unlike BREM-NET with nonlinear decoders, we also compare CEBRA with an ablation of BREM-NET in which encoders are kept nonlinear and decoder mappings are linear. As shown in Table A.8, this variant of BREM-NET still outperforms CEBRA in behavior decoding and multimodal neural prediction accuracy, confirming that BREM-NET's advantage arises from its disentangled latent dynamical formulation and multi-stage learning rather than nonlinear decoder alone.

Table A.8: **BREM-NET outperforms CEBRA even with linear decoders.**

| Models | Behavior Decoding (CC) | Neural prediction | |
| --- | --- | --- | --- |
| | | LFP (CC) | Spike (PP) |
| CEBRA | $0.5761 \pm 0.0380$ | $0.4225 \pm 0.0570$ | $0.3113 \pm 0.0164$ |
| BREM-NET w/ linear decoders | $0.6547 \pm 0.0341$ | $0.7975 \pm 0.0134$ | $0.3467 \pm 0.0023$ |

