# OpenReview forum: "Disentangling Behaviorally Relevant Latent Dynamics in Multimodal Neural Time-series"
_ICLR.cc/2026/Conference — Submitted to ICLR 2026_

### Official Review · Reviewer_5AY7 · 2025-10-28

**Soundness:** 3
**Presentation:** 3
**Contribution:** 3
**Rating:** 6
**Confidence:** 4

**Summary:**

The authors present BREM-NET, a multi-modal approach for modeling neural and behavioral activity. The approach additionally provides the ability to disentangle behaviorally-relevant neural dynamics from behavior-irrelevant dynamics. BREM-NET is applied to two datasets from monkey motor areas during different tasks, and benchmarked against several recent baselines.

**Strengths:**

BREM-NET is a clear, straightforward approach to nonlinearly modeling multi-modal neuro-behavioral datasets. It tackles a critical need for neuroscience with strong results on some preliminary datasets. Futhermore, the authors provide adequate benchmarking against existing models, and perform a thorough series of ablations to study the contributions from different model components. I would be very curious to see how this method applies to a growing number of neuro-behavioral mouse datasets, but the current work is strong enough as a proof of concept.

**Weaknesses:**

The paper is clearly written, except I found myself confused a few times in section 3 and figure 1. Figure 1, for example, contains a lot of information - not only the model architecture, but the three stage training approach. Somehow expanding the figure or otherwise rearranging the components or annotations could help readers understand these subtleties better.

**Questions:**

related work to consider including:
- Whiteway et al. Plos Comp Bio 2021: "Partitioning variability in animal behavioral videos using semi-supervised variational autoencoders"
- Wang et al. Neurips 2024: "Exploring Behavior-Relevant and Disentangled Neural Dynamics with Generative Diffusion Models"
- Shulz et al. Cell Reports 2025: "Modeling conditional distributions of neural and behavioral data with masked variational autoencoders"
- Zhang et al. ICML 2025: "Neural Encoding and Decoding at Scale"
- Wu et al. ICLR 2025: "Disentangling 3D Animal Pose Dynamics with Scrubbed Conditional Latent Variables"

L139: what does T ⊆ K mean? T and K are both integers. Does this mean that each value of t is also a value of k?

The definition of Z in L143 indicates the behavior must be sampled at the same time as the faster of the neural modalities. Is this in fact true? This seems like a strong limitation if so. Behavior may unfold much more slowly than the fastest neural modality, and behavior might be captured by, say, a video camera with its own particular framerate which cannot be synced to the neural recording apparatus (regardless of sampling rate).

In Eq 2, if one modality is unavailable at a given time step, the encoder output is set to zero. This seems suboptimal, because simply using zero is very different from properly indicating missing data. I would imagine you could potentially get much better performance by just linearly interpolating the lower-resolution signal. However, Table A.6 shows otherwise. Why does zero-replacement work?

Is Eq 2 a more detailed description of x_{k+1} = F(x_{k}) in Eq 1? If not, how does A, Enc_y, Enc_s relate to F?

I had to read section 3.2 several times before truly understanding the different stages. The final paragraph of this section is actually very helpful, and might better serve the reader if placed in the first paragraph of 3.2.

Why is \lambda_k in Eq 3 a product of C^(1) and C^(2) and not their sum?

BREM-NET results in Table A.4 are not the same as those in Table 1 - why the discrepancy?

Table 1: I'm not sure how fair some of these comparisons are:
- MVAE doesn't have access to temporal data; while this is how the method works, it seems like it could also be used in a more "temporal" mode where windows of data rather than single time points are used as training examples
- CEBRA: only using a linear mapping from the latents to the behavior/neural activity isn't a fair comparison to the BREM-NET model that uses nonlinear mappings. A interesting control would be nonlinear encoders and linear decoders for BREM-NET, which would then properly directly compare the latents learned by those two models.

I think extending BREM-NET to study shared latent dynamics across multiple brains is an extremely exciting future direction.

For another future direction, I'm curious what the authors have to say about using this model for studying multi-region interactions; for example, finding neural activity that is shared across regions versus private to each region.

4.1.1, second paragraph: decoding behavior separately from x^1 and x^2 is a good way to assess disentanglement. The decoding CC for x^2, 0.3367, is higher than I expected. an the authors speculate why this is not closer to zero? Does it have to do with the power of C_z? Presumably the more powerful the behavior decoder, the less residual behavioral information remains in x^2. If you replaced x^2 with the ground truth behavior would the decoding CC for x^2 be closer to zero? Might be an interesting control.

---

> ### Author Response · Authors · 2025-11-19
>
> We thank the reviewer for their thoughtful assessment of our work. We greatly appreciate their recognition that BREM-NET is a “*clear*” approach to nonlinearly modeling multi-modal neuro-behavioral datasets and that it tackles a “*critical need for neuroscience with strong results on some preliminary datasets.*” We are also grateful for the acknowledgment that our benchmarking and ablation studies are “*adequate*” and “*thorough*”.
>
> ### *Apply BREM-NET to a Neuro-Behavioral Mouse Dataset:
>
> To address the reviewer’s curiosity about applying BREM-NET to **mouse** datasets, we conducted an additional analysis during the rebuttal period using the **Allen Brain Observatory Visual Coding dataset [1]**, which includes two-photon calcium imaging and Neuropixels recordings from **mouse** visual cortex during passive viewing of natural movies (30 s, 30 Hz).
>  We applied BREM-NET to decode movie frame IDs (0–900) and observed:
>
> |Models|Frame ID prediction MAE(lower better)|
> |-|-|
> |Single-modal (Neuropixel)|22.99|
> |Single-modal (Calcium imaging)|58.51|
> |BREM-NET|20.70|
>
> These results show that BREM-NET improves frame ID prediction by integrating calcium and Neuropixels data. This demonstrates our model’s applicability beyond LFP and spikes in motor tasks, generalizing effectively across distinct neural modalities (e.g., LFP, Spiking activity, and Calcium imaging), different brain regions (e.g., motor cortex, visual cortex), different behaviors (e.g., random grid reach, center-out reach,  passive viewing of a movie), and different species (e.g., monkey, mouse).
>
> [1] de Vries, Saskia EJ, et al. "A large-scale standardized physiological survey reveals functional organization of the mouse visual cortex." Nature neuroscience 23.1 (2020): 138-151.

---

> ### Author Response · Authors · 2025-11-19
>
> ### [W1] Figure 1 Clarity:
>
> We thank the reviewer for acknowledging that our paper is “*clearly written*”. We also appreciate their feedback on Figure 1. We will make changes to Figure 1 in our revised manuscript to help readers better understand our framework.

---

> ### Author Response · Authors · 2025-11-19
>
> ### [Q1] Additional Related Work:
>
> We appreciate the suggested references and have incorporated all of these into our introduction and related work section in the revised manuscript.

---

> ### Author Response · Authors · 2025-11-19
>
> ### [Q2]: $T$ and $K$ notation:
>
> We thank the reviewer for pointing out this typo in our notations of sets $T$ and $K$. We have corrected the notation accordingly in the revised manuscript and used $\mathcal{T}$ and $\mathcal{K}$ to denote the corresponding time index sets. As the reviewer noted, the intended meaning of $\mathcal{T} \subseteq \mathcal{K}$ is that the indices in the lower-resolution set $\mathcal{T}$ form a subset of those in the higher-resolution set $\mathcal{K}$.

---

> ### Author Response · Authors · 2025-11-19
>
> ### [Q3]: Behavior Sampling rate:
>
> Behavior does **not** need to be sampled at the same resolution as the faster neural modality. In the datasets we evaluated on, behavior is available at the same time resolution as the faster time-scale neural activity. However, BREM-NET supports asynchronous sampling by performing latent updates at the fast neural timescale. Since inferring latents does not require any behavioral data, behavior can be decoded at any rate desired from the inferred latents. For example, behavior can be decoded at the rate at which behavioral samples are available in the training set, which may be slower than the rate at which neural samples are obtained.

---

> ### Author Response · Authors · 2025-11-19
>
> ### [Q4]: Handling of Missing Modalities:
>
> We thank the reviewer for raising this important point. The “zeroing” mentioned in Eq. (2) does **not** mean that the raw neural or behavioral data are replaced with zeros. Instead, the zeroing is applied **after the modality-specific encoder**, which means that the missing modality does not contribute to updating the inferred latent, not that it contributes as a zero value—specifically, when a modality is missing at time step $k$, we set its **encoder output** (e.g., $\mathrm{Enc}_ y(y_ k)$) to zero. This is similar to how in the Kalman filter, a missing sample corresponds to an infinite observation noise covariance and thus a zero Kalman gain for that modality in inferring the latent. The one-step-ahead inference of the Kalman filter can be written as:
>
> $ \hat{x}_ {k|k} = \hat{x}_ {k|k-1} + K_k \left ( y_ k - C \hat{x}_ {k|k-1} \right ) $
>
> where $K$ denotes the Kalman gain. For missing samples, the appropriate dimensions of the Kalman gain become zero, indicating a high degree of uncertainty in the observed signal [1, 2]. This means the missing modality does not update/correct the inferred latent.
> Similarly, our modality-specific encoder outputs can be seen as corrections/updates from the observed signals, which are set to zero in case observations are missing.
>
> In implementation, we define modality-specific binary masks that indicate the availability of each modality at each time step; these masks gate both the encoder outputs and their corresponding losses. This “zeroing” procedure allows the model to update/correct its latents using whatever subset of modalities is present, while ensuring that missing modalities do not update or distort the inferred latents.
>
> [1] Brookner, Eli. "Tracking and Kalman filtering made easy." (1998): 0471224197.
>
> [2] Maybeck, Peter S. Stochastic models, estimation, and control. Vol. 3. Academic press, 1982.

---

> ### Author Response · Authors · 2025-11-19
>
> ### [Q5]: Relation between Eq.1 and Eq.2:
>
> Eq. (2) provides the explicit parameterization of the latent transition function introduced abstractly in Eq. (3) as noted in lines 222–223 of the manuscript. In our formulation, Eq. (1) defines the *generative process* of the multimodal system—specifying how the latent state evolves in time through the recursion function \(F(\cdot)\) and how neural and behavioral observations ($y_t$, $s_k$, $z_t$) are generated from these latents.

---

> ### Author Response · Authors · 2025-11-19
>
> ### [Q6]: Summarizing paragraph added to beginning of Section 3.2 as suggested by the reviewer:
>
> We thank the reviewer for this helpful suggestion. We followed the reviewer's suggestion and added our final summarizing paragraph to the beginning of section 3.2 before moving into the details of each stage to improve readability.

---

> ### Author Response · Authors · 2025-11-19
>
> ### [Q7]: Product in $\lambda_k$ Calculation:
>
> In the Poisson point process model, the log-firing rate is related to the function of covariates–here the latents $x_k^{(1)​}$ and $x_k^{(2)​}$ [1]. Then the exponential of this function is taken as the firing rate to enforce non-negativity of the firing rates as detailed in Appendix A.1.1, lines 684-694. Thus, our Poisson decoders, $\( {C}_ \mathbf{s}^{(1)}(\cdot)\)$ and $\( {C}_ \mathbf{s}^{(2)}(\cdot)\)$, employ an $\(\exp(.)\)$ function and, as such, their output is multiplied, which is equivalent to adding the two log-firing rates in the Poisson point process model as shown in Appendix A.1.1, Equation 10.
>
> [1] Truccolo, Wilson, et al. "A point process framework for relating neural spiking activity to spiking history, neural ensemble, and extrinsic covariate effects." Journal of neurophysiology 93.2 (2005): 1074-1089.

---

> ### Author Response · Authors · 2025-11-19
>
> ### [Q8]: Clarification on Table 1 and Table A.4:
>
> The small numerical differences between Table 1 and Table A.4 arise from different random seeds used across runs.

---

> ### Author Response · Authors · 2025-11-19
>
> ### [Q9]: Fairness of Comparisons with CEBRA and MVAE:
>
> We thank the reviewer for the suggestion to match decoder complexity for fairer comparison with CEBRA. In CEBRA, the latents are mapped to behavior and neural activity with a linear projection layer. Following the reviewer comment, we now also compare CEBRA with an ablation of BREM-NET in which encoders are kept nonlinear and decoder mappings are linear. As shown in Table below, this variant of our method still outperforms CEBRA in behavior decoding and multimodal neural prediction accuracy, confirming that BREM-NET’s advantage arises from its disentangled latent dynamical formulation and multi-stage learning rather than nonlinear decoder alone.
>
> |Model|Behavior decoding (CC)|LFP prediction (CC)|Spike prediction (PP)|
> |-|-|-|-|
> |CEBRA|0.5761 $\pm$ 0.0380|0.4225 $\pm$ 0.0570|0.3113 $\pm$ 0.0164|
> |BREM-NET w/ linear decoders|0.6547 $\pm$ 0.0341|0.7975 $\pm$ 0.0134 |0.3467 $\pm$ 0.0023|
>
> Regarding MVAE, since we already have compared with dynamic baselines such as mmPLRNN and MMGPVAE, the only reason we compared with MVAE as well was to highlight the importance of temporal modeling in our methods and the dynamical baselines—demonstrating that models without explicit dynamics cannot capture cross-temporal neural–behavioral dependencies. Therefore, we reported the results using MVAE’s standard configuration. Our main baselines, MMGPVAE and mmPLRNN, however do include temporal dynamical modeling.

---

> ### Author Response · Authors · 2025-11-19
>
> ### [Q10]: Future Directions:  Multi-Brain and Multi-Region Interactions:
>
> We thank the reviewer for acknowledging that studying “*shared latent dynamics across multiple brains is an extremely exciting future direction.*”, which we note in our discussions. We also agree that extending BREM-NET to study multi-region neural interactions is indeed an exciting future direction. Conceptually, BREM-NET can be generalized to multi-region modeling by introducing a shared global latent $x^{(1)}$ that captures cross-region dynamics, while assigning region-specific latent subspace $x^{(2)}$ for the dynamics that are not shared across regions. The shared latent would represent population-level coordination—analogous to a communication subspace—while region-specific latents would capture local dynamics unique to each area. For example, applying this to multi-area motor and premotor recordings could reveal how preparatory signals in one region modulate activity in another. We have now added this to our discussion section.

---

> ### Author Response · Authors · 2025-11-19
>
> ### [Q11]: Decoding Behavior using $x^{(1)}$ or $x^{(2)}$:
>
> We thank the reviewer for this thoughtful question. To further assess the behavioral content of $x^{(2)}$, conditioned on $x^{(1)}$, we did a new analysis during the rebuttals period and computed behavior decoding CC by adding neural-specific latent states, $x^{(2)}$, to behaviorally-relevant latent states, $x^{(1)}$. As shown in Figure A.6 (a), the decoding accuracy saturates at 16 dimensions, which is the dimensionality of $x^{(1)}$, indicating that neural-specific latent states, $x^{(2)}$, do not encode unique behaviorally relevant information beyond what was explained in Stage 1. Together, the facts that $x^{(2)}$ behavior decoding is substantially poorer than $x^{(1)}$ (0.7629 vs. 0.3367), and 2) the fact that $x^{(2)}$ adds 0 decoding CC increase over $x^{(1)}$ show that the method has summarized the behaviorally relevant information in $x^{(1)}$. Indeed, this further suggests that $x^{(2)}$ and behavior signals share no mutual information conditioned on $x^{(1)}$. The small decoding CC of $x^{(2)}$ then indicates that there is some small residual information about behavior that is redundant with that in $x^{(1)}$ and can for example be due to upstream behavior-related input into the overall neuronal population that is not dissociatable without measuring the upstream regions.

---

> > ### Comment · Reviewer_5AY7 · 2025-11-23
> >
> > I thank the reviewers for their thorough answers to my questions. My final comment is, when the reviewers upload their next manuscript version can they indicate changes from the initial submission, by highlighting or coloring text?

---

> ### Author Response · Authors · 2025-11-25
>
> We thank the reviewer again for their detailed and constructive feedback throughout the review process. We have highlighted the changed parts of the manuscript in blue text, including addition in response to the reviewer as follows:
>
> - Figure 1 on **page 4** (W1): We revised figure 1 to make it easier to follow (by rearranging elements and writing a short description of learning stages).
> - Related works in page 2 (Q1): All related works proposed by the reviewer have been added to the introduction and related work section.
> - Method Section 3.2 **page 5** (Q6): We added  a summarizing paragraph to the beginning of Section 3.2 to help the reader understand the method better before going into the details of each stage.
> - New analysis on **page 30** (Q9): We added the new analysis of comparing BREM-NET w/  linear decoders to CEBRA as explained under Q9 into the appendix of our manuscript.
> - Multi-region as a future direction in **page 10** (Q10): We added a discussion of multi-region extension of BREM-NET in the discussion section.
> - New analysis on **page 29** (Q11): To quantify the behavioral information in the neural-specific latent $x^{(2)}$, we added the additional analysis as explained under Q11 into the appendix of our manuscript.
>
> We greatly appreciate the reviewer’s thoughtful engagement and detailed suggestions, all of which have substantively improved the clarity and strength of our work.

---

### Official Review · Reviewer_tDC9 · 2025-10-29

**Soundness:** 3
**Presentation:** 4
**Contribution:** 3
**Rating:** 8
**Confidence:** 4

**Summary:**

The paper proposes a nonlinear dynamical model, BREM-NET, that utilizes multiple neural modalities with behavioral data into a unified framework. The model can disentangle the latents into behaviorally relevant and neural-specific dynamics in multimodal neural time-series, thus making BREM-NET a more interpretable framework. The authors demonstrate the utility of BREM-NET in two multi-modal datasets.

**Strengths:**

- BREM-NET addresses an important issue of handling data that are not in sync, which is often the case for neuroscientific data.

- The literature review was very thorough, and the model comparisons were with some of the most relevant and novel models, which made the results even more compelling to me.

- The presentation was clear.

**Weaknesses:**

Although promising, I’m a little concerned with the optimization here, and its ability to correctly identify the disentangled latents. For example, for missing data, the authors state that the corresponding encoder output can be set to zero, but the different modalities can have different/finer scales so wouldn’t this cause issues for decoding? Also, are there any explicit constraints that would enforce private/shared latents in the model formulation other than the multi-step training procedure ?

**Questions:**

- Have the authors considered another more principled method of picking the shared latent dimensionality (along the lines of Giaffar et al., 2024)  as a starting point?

- How informative are the included latents? (e.g. the authors state that they fixed the total latent dimensionality for the experiments, but of these 64 total latents, what was the amount of information explained by each latent?)

- How does this compare to a model like BRAID (Vahidi et al., 2025), which also uses nonlinear dynamical modeling to  disentangle the shared dynamics between modalities?


Giaffar, H., Buxó, C. R., & Aoi, M. (2024, April). The Effective Number of Shared Dimensions Between Paired Datasets. In International Conference on Artificial Intelligence and Statistics(pp. 4249-4257). PMLR

Vahidi, P., Sani, O. G., & Shanechi, M. M. (2025). BRAID: input-driven nonlinear dynamical modeling of neural-behavioral data. arXiv preprint arXiv:2509.18627.

---

> ### Author Response · Authors · 2025-11-19
>
> We thank the reviewer for their thoughtful feedback. We greatly appreciate their recognition that our model addresses an “*important*” issue of handling asynchronous data and finding our literature review  “*very thorough”*  and our model comparisons and results “*compelling*” with “*clear*” presentation.
>
> ### [W1,W3]: Multi-stage Learning and Distinct Optimizations to enforce disentangled latents:
> > W1: Although promising, I’m a little concerned with the optimization here, and its ability to correctly identify the disentangled latents.
>
> > W3: Also, are there any explicit constraints that would enforce private/shared latents in the model formulation other than the multi-step training procedure ?
>
> We thank the reviewer for raising this important point. The reviewer’s understanding is indeed correct: the disentanglement is not enforced through explicit constraints or regularizations (such as minimizing the covariance between two sets of latent states), but through **distinct optimization objectives across three training stages**, each targeting a different subset of latents.
>
> Details of the method are in Section 3.2 and Appendix A.1.2. Briefly, BREM-NET achieves disentanglement through a **staged** and **conditioned** learning procedure. We learn two sets of neural latents, where the first set of neural latents $\mathbf{x}^{(1)}$ is learned purely by optimizing behavior decoding from multimodal neural modalities, and the second set of latents $\mathbf{x}^{(2)}$ is learned by optimizing neural reconstruction conditioned on the frozen first set of latents. By conditioning on $\mathbf{x}^{(1)}$, Stage 3 ensures that the new latents $\mathbf{x}^{(2)}$ learn to reconstruct only the part of neural activity that remains unexplained from $\mathbf{x}^{(1)}$. Note, all of these latents are inferred from neural activity alone and thus disentangle the neural variability into behaviorally relevant vs neural-specific variability.
>
> This hierarchical setup—learning $\mathbf{x}^{(1)}$ for behaviorally relevant neural variance and $\mathbf{x}^{(2)}$ for residual neural-specific variance conditioned on $\mathbf{x}^{(1)}$—is what enables BREM-NET to disentangle behaviorally relevant and neural-specific latent states within a unified nonlinear dynamical framework.
>
>
> Our **qualitative** and **quantitative empirical** analyses further support effective disentanglement:
> - Figure 2f shows that the behaviorally relevant neural latents $x^{(1)}$ and the neural-specific latents $x^{(2)}$ capture distinct dynamics and $x^{(1)}$ is closely aligned with the true behavior unlike $x^{(2)}$.
> - Figure A.3 shows that behavior decoding is much more accurate using $x^{(1)}$ compared to $x^{(2)}$, while neural predictions improve with $x^({2})$.
>
> Finally, to further address the reviewer’s concern, we conducted an additional analysis during the rebuttal period. We trained a nonlinear decoder (MLP) using all 64 latent dimensions jointly—i.e., both behavior-relevant $x^{(1)}$ and neural-specific $x^{(2)}$ latents—to predict behavior (note that BREM-NET decodes behavior only from behaviorally-relevant neural latents, which were chosen to be 16-dimensional). The performance of behavior decoding (CC) using both $x^{(1)}$ and $x^{(2)}$  was not significantly improved compared to using the behavior-relevant neural latents $x^{(1)}$ alone (0.7633 $\pm$ 0.0361 vs. 0.7629 $\pm$ 0.0256, $p=0.48$ one-sided Wilcoxon signed-rank test) as shown in Figure A.6 (a) in the revised manuscript. This result shows that including $x^{(2)}$ in addition to $x^{(1)}$ results in **no** improvement in behavior decoding performance, thus demonstrating that behaviorally relevant neural information is captured through $x^{(1)}$ and further evidencing disentanglement.
> Together, these theoretical and empirical results confirm that BREM-NET’s multi-stage optimization with distinct objectives reliably disentangles behaviorally relevant neural dynamics and neural-specific dynamics, without additional constraints.

---

> ### Author Response · Authors · 2025-11-19
>
> ### [W2]: Missing data handling:
>
> > W2: For example, for missing data, the authors state that the corresponding encoder output can be set to zero, but the different modalities can have different/finer scales so wouldn’t this cause issues for decoding?
>
> We thank the reviewer for raising this important point. The “zeroing” mentioned in Eq. (2) does **not** mean that the raw neural or behavioral data are replaced with zeros. Instead, the zeroing is applied **after the modality-specific encoder**, which means that the missing modality does not contribute to updating the inferred latent, not that it contributes as a zero value—specifically, when a modality is missing at time step $k$, we set its **encoder output** (e.g., $\mathrm{Enc}_y(y_k)$) to zero. This is similar to how in the Kalman filter, a missing sample corresponds to an infinite observation noise covariance and thus a zero Kalman gain for that modality in updating the inferred latent. For example, the one-step-ahead inference of the Kalman filter can be written as:
>
> $\hat{x}_ {k|k} = \hat{x}_ {k|k-1} + K_k \left ( y_k - C \hat{x}_{k|k-1} \right ) $
>
> where $K$ denotes the Kalman gain. For missing samples (infinite observation noise), the appropriate dimensions of the Kalman gain become zero, indicating a high degree of uncertainty in the observed signal [1, 2]. Our modality-specific encoder outputs can be seen as updates/corrections to the latents based on the observed signals, which are set to zero in case observations are missing.
> In implementation, we define modality-specific binary masks that indicate the availability of each modality at each time step; these masks gate both the encoder outputs and their corresponding losses. This “zeroing” procedure allows the model to update its latent dynamics using whatever subset of modalities is present, while ensuring that missing modalities do not update or distort the inferred latents.
>
> [1] Brookner, Eli. "Tracking and Kalman filtering made easy." (1998): 0471224197.
> [2] Maybeck, Peter S. Stochastic models, estimation, and control. Vol. 3. Academic press, 1982.

---

> ### Author Response · Authors · 2025-11-19
>
> ### [Q1]: Choice of Shared Latent Dimensionality:
>
> We thank the reviewer for this valuable suggestion. Incorporating data-driven statistical methods such as Giaffar et al. (2024) for determining shared latent dimensionality is indeed an excellent future direction. We have cited the work and note this point in the discussion of our revised manuscript as a promising approach for principled hyperparameter selection to further reduce the dimensionality of the hyperparameter search space. In the present study, we selected the shared behaviorally relevant latent dimensionality empirically based on behavior decoding performance in a validation set on one session from each dataset. As shown in Figure A.5 of manuscript, behavior decoding performance (CC) peaks around $n_1 = 16$, which is therefore selected as the number of behaviorally relevant latent dimensions.

---

> ### Author Response · Authors · 2025-11-19
>
> ### [Q2]: Informativeness of Latent States:
>
> We thank the reviewer for this excellent question. To directly quantify how much information each of the 64 latent dimensions carries about behavior and neural activity, we conducted a new latent–informativeness analysis during the rebuttal period.
>
> We constructed cumulative sets of latents by adding latents one by one to the sets, e.g., set i includes latents {1,..., i}, with latents 1,..., 16 being the behaviorally relevant neural latents and latents 17,...,64 being the neural-specific ones. We then trained nonlinear MLP decoders on each set of latents and evaluated the behavior decoding performance and spike and LFP prediction performances using those latents. Plotting each of these performances vs. the number of latents gave us the cumulative informativeness plots provided in **new Figure A.6 in the revised manuscript**. This figure shows that:
>
>
> 1. **Behavior information is highly concentrated in latents learned in Stage 1**
>  The cumulative behavior informativeness curve rises sharply within the first $ \sim 10–16$ latent dimensions and then saturates (Figure A.6 (a)). Adding the remaining latents (up to 64 total) provides no additional improvement in behavior decoding, indicating that behaviorally relevant information in neural data is captured by the behaviorally-relevant neural latents in Stage 1.
>
> 2. **Neural informativeness accumulates gradually across all 64 latents across Stage 1 and Stage 3**
> For neural activity, informativeness increases steadily as more neural-specific latents are included (Figure A.6 (b, c)). This demonstrates that neural-specific latents in Stage 3 capture distinct residual neural variability compared to behaviorally relevant neural latents in Stage 1.
>
> We have added this analysis and figure to the revised manuscript.

---

> ### Author Response · Authors · 2025-11-19
>
> ### [Q3]: Comparison with BRAID (Vahidi et al., 2025):
>
> We thank the reviewer for highlighting BRAID (Vahidi et al., 2025). BRAID focuses on disentangling *input-driven* versus *intrinsic* dynamics within a *single* neural modality jointly with behavior. Essentially it separates the intrinsic part of behaviorally relevant neural dynamics in a single neural modality and does not address multimodal neural data. In contrast, BREM-NET is designed for multimodal neural data and disentangles the behaviorally relevant dynamics in multimodal neural modalities from their neural-specific dynamics.
>
> In our framework, we introduce:
>
> - modality-specific encoders/decoders to handle heterogeneous statistical characteristics and different temporal resolutions (Poisson spikes with faster timescale vs Gaussian LFPs with slower timescale);
> - A three-stage learning scheme that explicitly disentangles behaviorally relevant versus neural-specific dynamics from multimodal neural modalities; and
> - The ability to fuse asynchronous or partially missing modalities.
>
> Since 1) BRAID’s focus is on modeling extrinsic inputs, 2) BRAID can only model single-modal Gaussian neural activity, and 3) BRAID cannot handle asynchronous multimodal neural data, it does not serve as a baseline for multimodal neural modeling.

---

### Official Review · Reviewer_AQP8 · 2025-10-31

**Soundness:** 3
**Presentation:** 3
**Contribution:** 3
**Rating:** 6
**Confidence:** 3

**Summary:**

Neural modalities have behavioral and neural dynamics. Disentangling these 2 types of dynamics is important for neural behavior relationships. While previous work has shown that disentangling behaviourally relevant neural dynamics enables better and more interpretable analysis of how neural activity relates to behavior, these methods are constrained to using a single neural modality like spikes alone or LFP alone. Therefore, the challenge is to capture both the fine scale of individual spikes and the larger network level scale of LFP.

The authors introduce BREM-NET: a non-linear dynamical model that can model how brain and behavior signals evolve over time, capturing complex nonlinear relationships. It can combine different kinds of brain and behavior data, handle different distributions and work with missing chunks of data too.

The model has 3 stages:
Stage 1 encodes 2 separate neural modalities, through 2 separate encoders (Enc_1 and Enc_2) and fuses them with a linear layer (concat the encodings and apply a linear projection). The output from this linear layer is then processed through an RNN that produces a latent state x_k1 which represents a time-evolving summary of all neural modalities up to time k.

The behavioural decoder (C) takes the latent signal x_k1 and passes it through a decoder to get the behaviour z_k. The loss function is the negative log likelihood (NLL) to predict behavior from behaviorally relevant states x_k1.

Stage 2 has 2 decoders specific to modality. Their task is to reconstruct the original data (spikes or LFP). NLL is used here too.

Stage 3, similar to stage 1, takes in the raw data + the learnt latent x_k1. An RNN learns a new set of latents x_k2. Stage 3 learning is unsupervised.

Monkey is doing a 2d cursor moving task. behavior - velocity of the cursor and neural data LFP. It merges the 2 modalities and tries to find the LFP data that is ‘causing’ the velocity data to change. Once that's done, it disentangles the remaining neural data that's not causing the behavior to change.

Result: the model can separate out the neural signals that cause the behavior better than base models. Behavior decoding is the most impressive jump. Other LFPcc and SpikePP are not very impressive but are at par with baseline.

**Strengths:**

The authors offer a novel method for optimally integrating information from multiple modalities. This has the potential to be valuable to the community, as simultaneous acquisition of data from multiple modalities, particularly single cell electrophysiology and LFP, is very common.

**Weaknesses:**

Although the summary tables benchmarking the new method against previousl methods shows impresive SOTA performance, it would help greatly to have more visuals and analysis comparing both when and why the new method performs better. For example, the only figure I see that shows this comparison is A4, wherein mmPLRNN looks like it's performing as well as the new method.

**Questions:**

1. In what way does the new method outperform old methods? In addition to the summary performance tables, please include plots that demonstrate where previous methods fail and this one succeeds.
2. In stage 3 of the model, it is not clear what is truly blind to the behavior - x_k1 comes from behavior, but the authors claim this stage is blind to behavior.

---

> ### Author Response · Authors · 2025-11-19
>
> We thank the reviewer for their thoughtful review. We greatly appreciate their comprehensive and insightful summary of our work. We are also grateful for their recognition of our model as a “*novel method for optimally integrating information from multiple modalities*” and for acknowledging its “*valuable to the community.*”
>
> ### [W1, Q1]: How BREM-NET outperforms existing methods:
>
> We thank the reviewer for this insightful question. To clarify where and why BREM-NET outperforms prior methods and what distinguishes its contributions beyond existing baselines, we summarize the key factors below:
>
> 1. **Multi-stage learning with behavior supervision in Stage 1**
> The multi-stage learning approach of BREM-NET is what allows it to not only achieve high behavior decoding but also achieve high neural prediction. In stage 1, BREM-NET learns the behaviorally relevant neural dynamics through a supervised behavior prediction objective. Then in stage 3, it learns the residual neural dynamics. Unlike BREM-NET, prior multimodal dynamical models such as mmPLRNN and MMGPVAE are *unsupervised* and lack explicit behavior supervision, as summarized in Table 1 and Appendix A.2.8–A.2.9. For fair comparison, we developed *supervised* extensions of these models by introducing an additional behavior prediction objective during training. Figure A.4, previously showed only the latents for *supervised* variants of mmPLRNN and MMGPVAE that we developed, and furthermore did not show the neural prediction performance. Now Figure A.4 (updated in the revised manuscript) includes both the original *unsupervised* and *supervised* latents of mmPLRNN and MMGPVAE. The updated Figure A.4 taken together with Table 1 highlight that:
>
>     - Unsupervised mmPLRNN and MMGPVAE achieve moderate neural prediction but poor behavior decoding due to missing behavioral guidance during learning.
>
>     - Supervised mmPLRNN and MMGPVAE improve behavior decoding but degrade neural prediction.
>
>     - In contrast, the multi-stage learning and disentanglement approach of BREM-NET allows it to learn both the behaviorally relevant latents $x^{(1)}$ and the neural-specific latents $x^{(2)}$, leading to strong performance in both behavior decoding and neural prediction—something no other model achieves simultaneously.
>
>
> 2. **Objective differences (ELBO vs. one-step-ahead prediction)**
> Second, mmPLRNN and MMGPVAE are trained to optimize ELBO, which can be a challenging objective due to KL-divergence, e.g., specific KL-term scaling procedures are developed to mitigate this problem. Unlike these models, BREM-NET is trained to optimize one-step-ahead prediction, which we believe can be another important contributing factor to the performance difference between BREM-NET and mmPLRNN/MMGPVAE.
>
> Beyond mmPLRNN/MMGPVAE, all the other baselines also lack the multi-stage learning. Furthermore, MVAE lacks modeling of temporal dynamics, MSID is linear and unsupervised, and CEBRA lacks explicit recursive modeling of temporal dynamics.

---

> ### Author Response · Authors · 2025-11-19
>
> ### [Q2]: Latents Are Not Inferred From Behavior:
>
> We thank the reviewer for raising this question. First, we clarify that $\mathbf{x}^{(1)}$ is not inferred from behavior but rather from multimodal neural activity by the RNN in Stage 1 $f^{(1)}(\cdot)$ (see Figure 1). Indeed, behavior information is not used to infer any of the latents. The behaviorally relevant latents $\mathbf{x}^{(1)}$ are thus still neural latents: they are the subset of neural latents that are behaviorally relevant.
>
> Once these behaviorally relevant neural latents are inferred, they are frozen and used in Stage 3 to compute the residual neural variability, which is the neural variability not already predictable by the behaviorally relevant neural latents. Then $\mathbf{x}^{(2)}$ are inferred from this residual neural variability. We emphasize that the only way behavior is ever used is during the model training by providing the loss function for stage 1. Behavior is never used to infer any of the latents, whether during training or afterwards during inference with the trained model.

---

### Official Review · Reviewer_y9c8 · 2025-11-04

**Soundness:** 1
**Presentation:** 3
**Contribution:** 1
**Rating:** 0
**Confidence:** 4

**Summary:**

The paper presents a model meant to disentangle neural activity and behavior using a model they cal BREM-NET. The feature that distinguishes their model for existing models is that it utilizes multiple neural modalities simultaneously while extracting latent variability that explains the behavior. The authors conduct a wide array of simulated experiments and real data validations.

**Strengths:**

The paper is mostly clearly written (with some exceptions that I point out below) and easy to follow. The benchmarking results look strong compared to competing methods but it is unclear what the value of this method is.

**Weaknesses:**

The primary, and unfortunately fatal, weakness of the paper is that the model does not disentangle latent variability. This is evident from both their model structure and their benchmarks. First, the behavioral latents are learned exclusively by supervised learning, making stage 1 of the model training essentially like training a nonlinear regression model with a low-dimensional bottleneck. However, the original neural activity AND the behavioral latents are mixed to learn the neural latents. So the neural latents cannot be regarded as residual variability that does not include behavioral information, i.e. neural latents are not disentangled from the behavioral latents. Indeed the behavioral latents are learned with no knowledge of the neural latents whatsoever. Moreover, in Section 4.1.2 (lines 401-402) the authors point out that U-BREM-NET matched BREM-NET on neural prediction, indicating a questionable role for $x^{(2)}_k$ beyond what U-BREM-NET could have provided. In essence, there is no disentanglement beyond what could have been achieved by Stage 1 alone.

Since the authors seem to couch the value of this model on its ability to "disentangle" as the title suggests, I cannot support acceptance of this submission.

**Questions:**

- The benchmarking procedures were not clear. Specifically, It is only in the appendix that it becomes clear that the authors were doing 1-step forward prediction to benchmark against competing baselines. However, for all but one of the models it is unclear how one-step forward prediction was conducted. It is clear for MMPLRNN, but for MMGPVAE the authors simply state "we obtained one-step-ahead latent states" but since MMGPVAE is not an explicit model of dynamics it is not clear precisely what they did to make such a prediction. For other models one-step-ahead prediction is not even mentioned.

---

> ### Author Response · Authors · 2025-11-19
>
> We thank the reviewer for their feedback. We appreciate their recognition of the paper being “*clearly written*” and “*easy to follow*” and their acknowledgement of our “*strong*” benchmarking results.
>
> ### [W1]: BREM-NET **does** disentangle latent variability:
>
> We thank the reviewer for the opportunity to clarify this fundamental point. We would like to clarify how BREM-NET explicitly disentangles latent variability in multimodal neural activity.
>
>  1. There seems to be a miscommunication about what behavioral latents are. The reviewer states that “However, the original neural activity AND the behavioral latents are mixed to learn the neural latents. So the neural latents cannot be regarded as residual variability that does not include behavioral information…” We clarify that behaviorally relevant latents are inferred, purely, from neural activity by the RNN in Stage 1 $f^{(1)}(\cdot)$. Critically, behavior information is not used at all to infer them or any of the latents. These behaviorally relevant latents are thus still neural latents: they are the subset of neural latents that are behaviorally relevant. Once these behaviorally relevant neural latents are inferred, they are frozen and used to compute the residual neural variability, which is the neural variability not already predictable by the behaviorally relevant neural latents. Indeed, computing the residual neural variability requires knowledge of what part of neural activity is behaviorally relevant and removing that part from the total variability. That is why behaviorally relevant neural latents are needed to compute this residual. Disentanglement is then achieved because the neural-specific latents are inferred from the residual neural variability, which no longer contains the behaviorally relevant information (captured through the behaviorally relevant neural latents). We realize that the terminology behaviorally relevant latents may have led to this miscommunication. To make it clear that behaviorally relevant latents are still latents in neural activity, we will call them behaviorally relevant neural latents.
>
> 2. The reviewer states “Indeed the behavioral latents are learned with no knowledge of the neural latents whatsoever.” As clarified above, behaviorally relevant latents are purely inferred from neural activity. They are indeed neural latents, they are those neural latents that are behaviorally relevant, i.e., can predict behavior from neural activity. We will refer to the behaviorally relevant latents as the behaviorally relevant neural latents for better clarity.
>
> We hope the above clarification shows how BREM-NET achieves disentanglement (Details of the method are in Section 3.2 and Appendix A.1.2.). Our **qualitative** and **quantitative empirical** analyses further support effective disentanglement:
> - Figure 2f shows that the behaviorally relevant neural latents $x^{(1)}$ and the neural-specific latents $x^{(2)}$ capture distinct dynamics and $x^{(1)}$ is closely aligned with the true behavior unlike $x^{(2)}$.
> - Figure A.3 shows that behavior decoding is much more accurate using $x^{(1)}$, compared to $x^{(2)}$ while neural predictions improve with $x^({2})$.
>
> Finally, to further address the reviewer’s concern, we conducted an additional analysis during the rebuttal period, showing that including $x^{(2)}$ in addition to $x^{(1)}$ results in **no** improvement in behavior decoding performance while significantly improving neural prediction; this demonstrates that behaviorally relevant information is captured through $x^{(1)}$ while residual neural-specific latents are captured through $x^{(2)}$, thus further evidencing disentanglement. Specifically, we trained a nonlinear decoder (MLP) using all 64 latent dimensions jointly—i.e., both behavior-relevant $x^{(1)}$ and neural-specific $x^{(2)}$ latents—to predict behavior (note that BREM-NET decodes behavior only from behaviorally-relevant neural latents, which were chosen to be 16-dimensional). The performance of behavior decoding (CC) using both $x^{(1)}$ and $x^{(2)}$  was not significantly improved compared to using the behavior-relevant neural latents $x^{(1)}$ alone (0.7633 $\pm$ 0.0361 vs. 0.7629 $\pm$ 0.0256, $p=0.48$ one-sided Wilcoxon signed-rank test) as shown in Figure A.6 (a) in the revised manuscript. In contrast, neural prediction improved significantly  when going from 16 latents to 64 latents in Figure A.6 (b,c). Together, these results indicate that behaviorally relevant information in neural activity is captured through $x^{(1)}$ while $x^{(2)}$ encodes neural-specific information that is not captured in $x^{(1)}$, thus evidencing the disentanglement of neural variability into the two subparts, behaviorally relevant neural latents and neural-specific latents.

---

> ### Author Response · Authors · 2025-11-19
>
> ### [W2]: Clarification on U-BREM-NET:
>
> The similar neural prediction accuracy between BREM-NET and its **unsupervised** variant (U-BREM-NET) is indeed expected and does not indicate a lack of disentanglement. As we described above, the behaviorally relevant neural latents are those neural latents that are behaviorally relevant. The neural-specific latents are those neural latents that are residual to the behaviorally relevant neural latents. As such, the union of the behaviorally relevant neural latents and the neural-specific latents should give us the overall predictable part of neural activity, which is indeed what U-BREM-NET aims to learn, though without any disentanglement. As such, it is expected that the union of the disentangled latents in BREM-NET and the non-disentangled latents in U-BREM-NET should get similar neural predictions. However, the behavior decoding of BREM-NET is substantially higher than U-BREM-NET because of the disentanglement capability afforded through multi-stage learning with distinct objectives as explained above and in manuscript.
> Indeed, U-BREM-NET can be viewed as a special case of BREM-NET where the number of behaviorally relevant latents $n_1 = 0$. It serves as an ablation to verify that **BREM-NET’s disentanglement does not compromise neural predictive power**. Instead, it allows us to disentangle the neural latents into behaviorally relevant vs. neural-specific, and improve behavior decoding through explicit behavioral supervision in stage 1 of learning when behaviorally relevant neural latents are being learned.

---

> ### Author Response · Authors · 2025-11-19
>
> ### [Q1]: Clarification on Benchmarking and One-Step Prediction Procedures:
>
> The one-step prediction procedures for each baseline are described in detail in the Appendix (A.2.5–A.2.9 in original manuscript),  and we will make sure to explicitly cross-reference these sections in the revised version for better visibility. To summarize, for MMGPVAE, we implemented one-step-ahead prediction following the same logic as for mmPLRNN, as detailed in Appendix A.2.9 page 19 of original manuscript. Specifically, since MMGPVAE does not include an explicit recurrent transition, following common practice in the Neural Latents Benchmark (NLB) [1], we inferred one-step-ahead latent states by zeroing neural inputs after timestep $k-1$ and passing only observations up to $k-1$ through the inference network, ensuring that the predicted latent at $k$ depends solely on past neural data. These predicted latents were then decoded to obtain one-step-ahead neural outputs.
>
> For MVAE, as detailed in in Appendix A.2.6 page 18, because the model is static, one-step-ahead prediction is not defined, and we therefore do not report neural prediction metrics for it, as noted in the original manuscript (Table 1). For MSID, as explained in Appendix A.2.5 page 18, this model naturally supports one-step prediction through its latent dynamics; thus, no modification was required from the original manuscript that reports one-step-ahead prediction.
>
> [1] Pei, Felix, et al. "Neural latents benchmark'21: evaluating latent variable models of neural population activity." arXiv preprint arXiv:2109.04463 (2021).

---

### Author Response · Authors · 2025-11-25
**Global Response**

We thank the reviewers for taking the time to review our submission and for providing constructive comments regarding our work. We are pleased that the reviewers found that our work is “*novel*”, “tackles a *critical need* for neuroscience”, and “has the potential to be *valuable* to the community”, and that they acknowledged that our benchmarking and ablation studies are “*strong*”, “*thorough*”, and “*compelling*”.

We have addressed all comments by performing several new analyses and providing clarifications/discussions. Our responses and results have been provided either inline in our individual responses to reviewers (e.g., as tables and text), and/or included as revisions in the manuscript. We have uploaded the revised manuscript, which now addresses all comments from all reviewers and whose revisions are highlighted in blue text. We once again thank all reviewers for the valuable and constructive feedback, which we believe has significantly improved our manuscript. Should there be any remaining questions or concerns, we would be happy to clarify them, perform additional analyses to address them, and engage in further discussion.

---

### Author Response · Authors · 2025-12-03
**Global Response**

We thank all reviewers for the thoughtful comments. We introduced BREM-NET to enable nonlinear multimodal neural fusion while also disentangling behaviorally relevant multimodal neural dynamics, even with asynchronous neural modalities, which is a major challenge in real-world neural recordings. As described in our detailed responses, we have fully addressed each of the reviewers’ points and updated the paper accordingly. Although the discussion period for ICLR closed early this year, before the reviewers had a chance to provide feedback, we hope the reviewers find that the revisions meaningfully strengthen the work.

---

### Meta-Review · Area_Chair_pC9b · 2026-01-08

**Summary:**

This paper develops BREM-NET, a model for multimodal neural decoding from spikes and LFP that aims to disentangle behaviorally relevant neural activity from non-behaviorally guided activity. They leverage an existing line of work where two sets of latents are learned, one set for supervised decoding of behavior and the other set to capture the remaining information in the neural activity. In this work, this separation is achieved through a multi-stage optimization procedure which first extracts latents to predict behavior, and then fits a set of decoders that can predict the residual multimodal neural activity not predicted by the neural activity that could be predicted from the behaviorally relevant neural latents. Their claim is that this disentangles information from the two sets of latents, one which contains information about the behavior and the other set of latents tell you about other information not related to behavior.

The authors make disentanglement a core component of their motivation and abstract, yet the demonstration of disentanglement in the main paper is relatively weak, consisting of visual inspection of the two sets of latents, as well as a result in the Appendix (A.3) which investigates the decoding of behavior or neural encoding in each set of latents. This starts to get at the question of disentanglement and should be highlighted in the main paper. Both disentanglement metrics and synthetic examples would likely be needed to appropriately make claims about disentanglement; there are examples of this being done in prior works like PI-VAE (Zhou, 2020) and MM-GPVAE (Gondur, 2023).

The improvements in behavior decoding are impressive, however, the performance from the U-BREM-NET, an unsupervised variant of their model trained without behavior supervision or disentanglement performs nearly as well on behavior decoding and outperforms the BREM-NET model on neural prediction. This suggests the multimodal encoder design is responsible for much of the performance. The multistep training seems to boost the decoding performance thanks to the explicit conditioning on the behavior task in Stage 1 of the model. The true benefit of disentanglement is thus further brought into question.

Both Reviewer y9c8 and tDC9 raised concerns about the extent to which the model truly disentangles the data, and if the proposed multi-stage "optimization procedure would correctly identify the disentangled latents” (tDC9). Reviewer y9c8 also raised the concern of the added value of disentanglement since the neural variability is better predicted by the unsupervised variant of the model.

During the rebuttal period, the authors provided further clarifications on the model structure and training, and conducted an additional analysis (A.5.8) of the latents. They also ran additional analysis on mouse datasets from Allen to show further utility of the approach based upon feedback from Reviewer 5AY7. Most reviewers appeared to be satisfied with the responses, but Reviewer y9c8 didn’t engage in the discussion before the portal was closed.

While there are some merits of the work, the claims about disentanglement have not been validated in a rigorous manner. The reviewers also pointed out similar concerns but their impact on their scores was highly variable. Based upon a careful reading through the paper and reviews, I believe that a lot of revisions must be made to the paper before it would be able to be published. Adding disentanglement analyses and metrics, as well as synthetic examples would further strengthen the work. If positioned instead as a multimodal neural decoder-encoder, further comparisons to state-of-the-art multimodal methods like NEDS (Zhao, 2025) would be also be useful.

**Reviewer Concerns:**

The reviewer's concerns were partially alleviated through further clarification of the model objective, along with Figure A3, and the new analysis of A.5.8. However, these results must be fully integrated into the main text, and additional metrics/examples must be provided to support claims about disentanglement. Disentanglement and claims should also be validated for the multi-session results which are now only based upon visual inspection.

**Reviewer Scores:**

The reviewers were likely to stick with their same scores.

---

### Decision · Program_Chairs · 2026-01-26

Reject